# Fair Scheduling for Time-dependent Resources

**Bo Li** [*]
Department of Computing
The Hong Kong Polytechnic University
Hung Hom, Hong Kong
`comp-bo.li@polyu.edu.hk`

**Minming Li** [*]
Department of Computer Science
City University of Hong Kong
Kowloon Tang, Hong Kong
`minming.li@cityu.edu.hk`

**Ruilong Zhang** [* †]
Department of Computer Science
City University of Hong Kong
Kowloon Tang, Hong Kong
`ruilzhang4-c@my.cityu.edu.hk`

## Abstract

We study a fair resource scheduling problem,nwhere a set of interval jobs are to be allocated to heterogeneous machines controlled by intellectual agents. Each job is associated with release time, deadline and processing time such that it can be processed if its complete processing period is between its release time and deadline. The machines gain possibly different utilities by processing different jobs, and all jobs assigned to the same machine should be processed without overlap. We consider two widely studied solution concepts, namely, maximin share fairness and envy-freeness. For both criteria, we discuss the extent to which fair allocations exist and present constant approximation algorithms for various settings.

## 1  Introduction

With the rapid progress of AI technologies, AI algorithms are widely deployed in many societal settings such as the distribution of job and education opportunities where complex social effects may significantly diminish the performance of the algorithms. To motivate our study, let us consider a problem faced by the Students Affairs Office (SAO). An SAO clerk is assigning multiple part-time jobs to the students who submitted job applications. Each part-time job occupies a consecutive time period within a possibly flexible interval. For example, an one-hour math tutorial needs to be given between 8:00am and 11:00am on June 26th. A *feasible* assignment requires that the jobs assigned to an applicant can be scheduled without mutual overlap. The students are heterogeneous, i.e., different students may hold different job preferences. It is important that the students are treated equally in terms of getting job opportunities, and thus the clerk's task is to make the assignment fair.

The SAO problem falls under the umbrella of the research on *job scheduling*, which has been studied in numerous fields, including operations research Gentner *et al.* [2004], machine learning Paleja *et al.* [2020], parallel computing Drozdowski [2009], cloud computing Al-Arasi and Saif [2020], etc. Following the convention of job scheduling research, each part-time job, or *job* for short, is associated with release time, deadline, and processing time. The students are modeled as *machines*, who have different utility gains for completing jobs. Traditionally, the objective of designing scheduling algorithms is solely focused on efficiency or profit. However, motivated by various real-world AI driven deployments where the data points of the algorithms are real human beings who should be

---

[*]Equal contribution
[†]Corresponding author

treated unbiasedly, addressing the individual fairness becomes important. Accordingly, the past several years has seen considerable efforts in developing fair AI algorithms Chierichetti *et al.* [2017], where combinatorial structures are incorporated into the design, such as vertex cover Rahmattalabi *et al.* [2019], facility location Chen *et al.* [2019] and knapsack Amanatidis *et al.* [2020].

It is noted that people have different criteria on evaluating fairness, and in this work, we consider two of the most widely accepted definitions. The first is motivated by the max-min objective, i.e., maximizing the worst-case utility, which has received observable attention for various learning scenarios Rahmattalabi *et al.* [2019]. However, for heterogeneous agents, optimizing the worst case is not enough, as different people have different perspectives and may not agree on the output. Accordingly, one popular research agenda is centered around computing an assignment such that everyone believes that it (approximately) maximizes the worst case utility. This criterion is named *maximin* share (MMS) fairness in Budish [2010]. The second one is *envy-freeness* (EF), which has been very widely studied in social sciences and economics but arguably less explored in machine learning. Informally, an assignment is called EF if everyone believes she has obtained the best resource compared with any other agent's assignment. We note that, due to the scheduling-feasible constraint, some jobs may not be allocated. Thus EF alone is not able to satisfy the agents as keeping all resources unallocated does not incur any envy among them, but the agents envy the charity where unallocated/disregarded items are assumed to be donated to a charity. To resolve this issue, in this work, we want to understand how we can compute allocations that are simultaneously EF and Pareto efficient (PO), where an allocation is called PO if there does not exist another allocation that makes nobody worse off but somebody strictly better off.

Recently, Chiarelli *et al.* [2020] and Hummel and Hetland [2021] studied the fair allocation of conflicting items, where the items are connected via graphs. An edge between two items means they are in conflict and should be allocated to different agents. However, in our model, the conflict among items cannot be described as the edges in a graph. For example, two one-hour tutorials between 9:00am and 11:00am can be feasibly scheduled, but three such tutorials are not feasible any more. For a comprehensive introduction to the various constraints, including conflict constraints, studied in fair division, we refer the interested reader to the recent survey of Suksompong [2021].

## 1.1 Main Results

We study the fair interval scheduling problem (FISP), where fairness is captured by MMS and EF. For each of them, we design approximation algorithms to compute MMS or EF1 schedules.

**Maximin Share.** Informally, a machine's MMS is defined to be her optimal worst-case utility in an imaginary experiment: she partitions the items into $m$ bundles but was the last to select one, where $m$ is the number of agents. It is noted that as the machines are heterogeneous, they may not have the same MMS value. Our task is to investigate the extent to which everyone agrees on the final allocation. A job assignment is called $\alpha$-approximate MMS fair if every machine's utility is no less than $\alpha$ fraction of her MMS value. Our main result in this part is an algorithmic framework which ensures a $1/3$-approximate MMS schedule, and thus improves the best known approximation of $1/5$ which is proved for a broader class of valuation functions – XOS Ghodsi *et al.* [2018]. Interestingly, in the independent and parallel work Hummel and Hetland [2021], the authors also show the existence of $1/3$-approximate MMS for graphically conflicting items. With XOS valuation oracles, Ghodsi *et al.* [2018] also designed a polynomial-time algorithm to compute a $0.125$-approximate MMS allocation. As a comparison, by slightly modifying our algorithm, it returns a $0.24$-approximate MMS allocation in polynomial time, without valuation oracles. When all jobs are rigid, i.e., processing time = deadline - release time, our problem degenerates to finding a partition of an interval graph such that the minimum weight of the independent set for each subgraph is maximized. Recently, a pseudo-polynomial-time algorithm is given in Chiarelli *et al.* [2020] for constant number of agents. In this sense, we generalize this problem to flexible jobs and design approximation algorithms for arbitrary number of agents.

**Main Result 1.** For an arbitrary FISP instance, there exists a $1/3$-approximate MMS schedule, and a $(0.24 - \epsilon)$-approximate MMS schedule can be found in polynomial time, for any constant $\epsilon > 0$.

**EF1+PO.** EF is actually a demanding fairness notion, in the sense that any approximation of EF is not compatible with PO. Instead, initiated by Lipton *et al.* [2004], most research is focused on its relaxation, *envy-freeness up to one item* (EF1), which means the envy between two agents may exist

but will disappear if some item is removed. Unfortunately, EF1 and PO are still not compatible even if all jobs are rigid and agents have unary valuations. However, the good news is, if all jobs have unit processing time, an EF1 and PO schedule is guaranteed to exist and can be found in polynomial time. This result continues to hold when agent valuations are weighted but identical. It is shown in Biswas and Barman [2018] that under laminar matroid constraint an EF1 and PO allocation exists when agents have identical utilities, but finding it may need exponential time. We improve this result in two perspectives. First, our feasibility constraints, even for unit jobs, are not necessarily laminar matroid. Second, our algorithm runs in polynomial time.

**Main Result 2.** No algorithm can return an EF1 and PO schedule for all FISP instances, even if all jobs are rigid and valuations are unary. When all jobs have unit processing time and valuations are (weighted) identical, an EF1 and PO schedule can be computed in polynomial time.

Although exact EF1 and PO are not compatible, we prove that for an arbitrary FISP instance, there always exists a $1/4$-approximate EF1 and PO schedule, which coincides with Wu *et al.* [2021]. If all jobs have unit processing time, a $1/2$-approximate EF1 and PO schedule exists. To prove this result, we consider Nash social welfare – the geometric mean of all machines' utilities. We show that a Nash social welfare maximizing schedule satisfies the desired approximation ratio. This result is in contrast to the corresponding one in Caragiannis *et al.* [2016], which shows that without any feasibility constraints, such an allocation is EF1 and PO. We also show that both approximations are tight.

**Main Result 3.** For any FISP instance, the schedule maximizing Nash social welfare is PO and $1/4$-approximate EF1. If all jobs have unit processing time, it is $1/2$-approximate EF1.

**EF1+IO** By above results, we observe that PO is too demanding to measure efficiency in our model. One milder requirement is *individual optimality* (IO). Intuitively, an allocation is called IO if every agent gets the best feasible subset of jobs from the union of her current jobs and unscheduled jobs. We show that EF1 is still not compatible with IO in the general case. But for unary valuations, we obtain positive results and design polynomial time algorithms for (1) computing an EF1 and IO schedule for rigid jobs, and (2) computing an EF1 and 1/2-approximate IO schedule for flexible jobs. To prove these results, we utilize two classic algorithms *Earliest Deadline First* and *Round-Robin*. We defer this part completely to the full version Li *et al.* [2021].

## 1.2 Other Related Works

Since computing feasible job sets to maximize the total weight is NP-hard Garey and Johnson [1979], various approximation algorithms have been proposed Bar-Noy *et al.* [2001]; Berman and DasGupta [2000]; Chuzhoy *et al.* [2006], and the best known approximation ratio is $0.644$ Im *et al.* [2020]. For rigid instances, the problem is polynomial-time solvable Schrijver [1999]. Recently, scheduling has been studied from the perspective of machine learning, including developing learning algorithms to empirically solve NP-hard scheduling problem Zhang *et al.* [2020]; Paleja *et al.* [2020], and predicting uncertain data in order to optimize the performance in the online setting Purohit *et al.* [2018]. Fairness has been concerned in the scheduling community in past decades Ajtai *et al.* [1998]; Baruah and Lin [1998]; Baruah [1995]. Most of these works aim at finding a fair schedule for the jobs, such as balancing the waiting and completion time Bilò *et al.* [2016]; Im and Moseley [2020].

MMS allocation for indivisible resources has been widely studied since Budish [2010]. Unfortunately, it is shown in Kurokawa *et al.* [2018]; Ghodsi *et al.* [2018]; Feige *et al.* [2021] that an exact MMS fair allocation may not exist. Thereafter, a string of approximation algorithms for various valuation types are proposed, such as additive Garg and Taki [2020], matroid-rank function Babaioff *et al.* [2021]; Barman and Verma [2021], submodular Barman and Krishnamurthy [2020]; Ghodsi *et al.* [2018], XOS and subadditive Ghodsi *et al.* [2018]. Regarding EF1, in the unconstrained setting, an allocation that is both EF1 and PO is guaranteed to exist Caragiannis *et al.* [2016]; Barman *et al.* [2018]. However, when there are constraints, such as cardinality and knapsack, the general compatibility is still open Biswas and Barman [2018, 2019]; Wu *et al.* [2021]; Gan *et al.* [2021]; Dror *et al.* [2021].

## 2 Preliminaries

### 2.1 Fair Interval Scheduling Problem

In a fair interval scheduling problem (FISP), we are given a job-machine system, which is denoted by tuple $(J, A, \mathbf{u}_A)$. $J = \{j_1, \cdots, j_n\}$ represents a set of $n$ jobs (also called resources or items) and $A = \{a_1, \cdots, a_m\}$ is a set of $m \geqslant 2$ machines controlled by agents. In this work, machines and agents are used interchangeably. We consider discrete time, and for $t \in \mathbb{N}_+$, let $[t, t+1)$ denote the $t$-th *time slot*. Each $j_i \in J$ is associated with release time $r_i \in \mathbb{N}_+$, deadline $d_i \in \mathbb{N}_+$, and processing time $p_j \in \mathbb{N}_+$ such that $p_i \leqslant d_i - r_i + 1$. The $[r_i, d_i]$ is called a job interval, which can also be viewed as a set of consecutive time slots, $\{r_i, r_i + 1, \cdots, d_i\}$. Job $j_i$ can be processed successfully if it is offered $p_i$ consecutive time slots within $[r_i, d_i]$. Each machine can process at most one job at each time slot and a set of jobs $J' \subseteq J$ is called *feasible* if all jobs in $J'$ can be processed without overlap on a single machine. For a job $j_k \in J$, agent $a_i \in A$ gains utility $u_i(\{j_k\}) \geqslant 0$ if $j_k$ is successfully processed by $a_i$. We slightly abuse the notation and assume that $u_i(j_k) = u_i(\{j_k\})$. We use $u_i$ to denote $a_i$'s utility function, and define $\mathbf{u}_A = (u_i)_{i \in A}$. For a feasible set of jobs $S$, the agent's utility is additive, i.e., $u_i(S) = \sum_{j_k \in S} u_i(j_k)$. For an arbitrary set of jobs that may not be feasible, the agent's utility is the maximum she can obtain by processing a feasible subset, i.e.,

$$u_i(S) = \max_{S' \subseteq S:\ S'\ \text{is feasible}} \sum_{j_k \in S'} u_i(j_k).$$

It is noted that $u_i(\cdot)$'s are not additive for infeasible set of jobs and the computation of its value is NP-hard Garey and Johnson [1979]. In the full version Li *et al.* [2021], we show that they are actually XOS, which is a special type of subadditive functions. We call these $u_i(\cdot)$'s *interval scheduling* (IS) functions.

A *schedule* or *allocation* $\mathbf{X} = (X_1, \cdots, X_m)$ is defined as an ordered partial partition of $J$, where $X_i$ is the jobs assigned to agent $a_i$, such that $X_i \cap X_j = \emptyset$ for $i \neq j$ and $X_1 \cup \cdots \cup X_m \subseteq J$. Let $X_0 = J \setminus \bigcup_{i \in [m]} X_i$ denote all unscheduled jobs, which is regarded as the donation to a *charity*. A schedule $\mathbf{X}$ is called *feasible* if $X_i$ is feasible for all $a_i \in A$, i.e., all jobs in $X_i$ can be successfully processed by $a_i$. Note that since jobs in $X_0$ are not scheduled, $X_0$ is not necessarily feasible. Observe that any infeasible schedule $\mathbf{X}$ is equivalent to a feasible schedule $\mathbf{X}'$ by setting each $X_i'$ to be the feasible subset of $X_i$ that maximizes $a_i$'s utility and $X_0' = J \setminus \bigcup_{i \in [m]} X_i'$. We call an instance *rigid* if $p_i = d_i - r_i + 1$, for all $j_i \in J$, i.e., the jobs need to occupy the entire job intervals. For rigid instances, the feasibility constraints can be described via interval graphs and the computation of $u_i(S)$ for any $S \subseteq J$ can be done in polynomial time Kleinberg and Tardos [2006].

### 2.2 Solution Concepts

We first define the maximin value for any utility function $u$, item set $S$ and the number of agents $k$. Let $\mathcal{F}(S, k)$ be the set of all $k$-partial-partitions of $S$ and

$$\mathsf{MMS}^u(S, k) = \max_{(S_1, \cdots, S_k) \in \mathcal{F}(S,k)} \min_{i \in [k]} u(X_i).$$

For any FISP instance $(J, A, \mathbf{u}_A)$ with $m = |A|$, agent $a_i \in A$'s maximin share (MMS) is given by

$$\mathsf{MMS}_i(J, m) = \mathsf{MMS}^{u_i}(J, m).$$

When the parameters are clear in the context, we write $\mathsf{MMS}_i = \mathsf{MMS}_i(J, m)$ for simplicity. If a schedule $\mathbf{X}$ achieves $\mathsf{MMS}_i$, i.e., $\min_{k \in [m]} u_i(X_k) = \mathsf{MMS}_i$, it is called an MMS schedule for $a_i$.

**Definition 1** ($\alpha$-MMS Schedule). *For $0 < \alpha \leqslant 1$, a schedule $\mathbf{X} = (X_1, \cdots, X_m)$ is called $\alpha$-approximate MMS ($\alpha$-MMS) if $u_i(X_i) \geqslant \alpha \cdot \mathsf{MMS}_i$. When $\alpha = 1$, $\mathbf{X}$ is called an MMS schedule.*

We next introduce envy freeness (EF). An EF schedule $\mathbf{X} = (X_1, \cdots, X_m)$ requires everybody's utility to be no less than her utility for any other agent's bundle, i.e., $u_i(X_i) \geqslant u_i(X_k)$ for any $a_i, a_k \in A$. Since EF is over demanding for indivisible items, following the convention of fair division literature, in this work, we mainly consider EF1.

**Definition 2** ($\alpha$-EF1 Schedule). *For $0 < \alpha \leqslant 1$, a schedule $\mathbf{X} = (X_1, \cdots, X_m)$ is called $\alpha$-approximate envy-free up to one item ($\alpha$-EF1) if for any two agents $a_i, a_k \in A$,*

$$u_i(X_i) \geqslant \alpha \cdot u_i(X_k \setminus \{j\}) \text{ for some } j \in X_k.$$

*When $\alpha = 1$, $\mathbf{X}$ is called an EF1 schedule.*

We observe that an *empty* schedule is trivially EF and EF1, i.e., $X_0 = J$ and $X_i = \emptyset$ for all $a_i \in A$. However, this is a highly inefficient schedule, and thus we also want the schedule to be Pareto optimal.

**Definition 3** (PO schedule). *A schedule $\mathbf{X} = (X_1, \cdots, X_m)$ is called Pareto Optimal (PO) if there does not exist an alternative schedule $\mathbf{X}' = (X_1', \cdots, X_m')$ such that $u_i(X_i') \geqslant u_i(X_i)$ for all $a_i \in A$, and $u_k(X_k') > u_k(X_k)$ for some $a_k \in A$.*

We note that any approximation of EF is not compatible with PO, even in the very simple setting with two machines and a single job. In the full version Li *et al.* [2021], we introduce another efficiency criterion, *individual optimality* (IO), which is weaker than PO and study the compatibility between EF1 and IO.

## 3 Approximately MMS Scheduling

Before introducing our algorithmic framework, we first recall the best known existential and computation results for MMS scheduling problems.

**Observation 1** (Ghodsi *et al.* [2018]). *For an arbitrary FISP instance, there exists a 1/5-MMS schedule and a 1/8-MMS schedule can be computed in polynomial time, given XOS function oracle.*

### 3.1 Algorithmic Framework

In this section, we present our algorithmic framework and prove that it ensures a $1/3$-MMS schedule. The algorithm has two parameters, a threshold vector $(\gamma_1, \cdots, \gamma_m)$ with $\gamma_i \geqslant 0$ and a $\beta$-approximation algorithm for IS functions, where $0 \leqslant \beta \leqslant 1$. In this section, we set $\gamma_i = \mathsf{MMS}_i$ for each $a_i \in A$. We can pretend that $\beta = 1$ to understand the existential result easily. Note that the computations of each $\mathsf{MMS}_i$ and exact value for IS functions are NP-hard, and in Section 3.2, we show how to gradually adjust the parameters to make it run in polynomial time. The high-level idea of the algorithm is to repeatedly fill a bag with unscheduled jobs (which may not be feasible) until some agent values it for no less than a threshold and takes away the bag. Then this agent reserves her best feasible subset of the bag, and returns the remaining jobs to the algorithm. By carefully designing the thresholds, we show that everybody can obtain at least $\frac{\beta}{\beta+2}$ of her MMS.

#### 3.1.1 Pre-processing

As we will see, the above bag-filling algorithm works well only if the jobs are small, i.e., $u_i(j_k) \leqslant \frac{\beta}{\beta+2} \cdot \gamma_i$ for all $a_i \in A$ and $j_k \in J$. We first introduce the following property, which is used to deal with large jobs. Intuitively, Lemma 1 implies that after allocating an arbitrary job to an arbitrary agent, the remaining agents' MMS values in the reduced sub-instance do not decrease. A similar result for additive valuations is proved in Amanatidis *et al.* [2017].

**Lemma 1.** *For any instance $\mathcal{I} = (J, A, \mathbf{u}_A)$ with $|A| = m$, the following inequality holds for any $a_i \in A$ and any $j_k \in J$,*

$$\mathsf{MMS}_i(J \setminus \{ j_k \}, m-1) \geqslant \mathsf{MMS}_i(J, m).$$

We use Lemma 1 to design Algorithm 1 which repeatedly allocates a large job to some agent and removes them from the instance until there is no large job.

---

**Algorithm 1.** Matching Procedure

---

**Input:** Arbitrary FISP instance $\mathcal{I} = (J, A, \mathbf{u}_A)$; Thresholds $(\gamma_1, \cdots, \gamma_m)$.
**Output:** (1) Sub-instance $\mathcal{I}' = (J', A', \mathbf{u}_{A'})$ such that $u_i(j_k) \leqslant \frac{\beta}{\beta+2} \cdot \gamma_i$ for all $a_i \in A'$ and $j_k \in J'$;
    (2) Partial Schedule $(X_r)_{a_r \in A \setminus A'}$.
 1: Initialize $A' = A$ and $J' = J$.
 2: **while** there is an agent $a_i \in A'$ and a job $j_k \in J'$ with $u_i(j_k) > \frac{\beta}{\beta+2} \cdot \gamma_i$ **do**
 3:      Set $X_i = \{ j_k \}$, $A' = A' \setminus \{ a_i \}$, and $J' = J' \setminus \{ j_k \}$.
 4: **end while**

---

By Lemma 1, it is straightforward to have the following lemma.

**Lemma 2.** *For any instance $\mathcal{I} = (J, A, \mathbf{u}_A)$ with $(\gamma_1, \cdots, \gamma_m)$, the partial schedule $(X_r)_{a_r \in A \setminus A'}$ and the reduced instance $\mathcal{I}' = (J', A', \mathbf{u}_{A'})$ returned by Algorithm 1 satisfy $u_r(X_r) \geqslant \frac{\beta}{\beta+2} \cdot \gamma_r$ for all $a_r \in A \setminus A'$ and $\mathsf{MMS}_i(J', |A'|) \geqslant \mathsf{MMS}_i(J, |A|)$ for all $a_i \in A'$.*

### 3.1.2 Bag-Filling Procedure

Let $\mathcal{I} = (J, A, \mathbf{u}_A)$ be an instance such that $|A| = m$ and $u_i(j_k) \leqslant \frac{\beta}{\beta+2} \cdot \gamma_i$ for all $a_i \in A$ and $j_k \in J$. We show the Bag-Filling Procedure in Algorithm 2, with parameters $(\gamma_1, \cdots, \gamma_m)$ and $\beta$-approximation algorithm for IS functions. For each $a_i \in A$, we use $u'_i : 2^J \to \mathbb{R}_+$ to denote the approximate utility, and thus $u'_i(S) \geqslant \beta \cdot u_i(S)$ for any $S \subseteq J$. Intuitively, it keeps a bag $B$ and repeatedly adds an unscheduled job into it until some agent $a_i$ first values this bag (under the approximate utility function $u'_i$) for at least $\frac{\beta}{\beta+2} \cdot \gamma_i$. If there are more than one such agents, arbitrarily select one of them. Then $a_i$ gets assigned a feasible subset $X_i \subseteq B$ with $\sum_{j_l \in X_i} u_i(j_l) = u'_i(B)$, and returns $B \setminus X_i$ to the algorithm. This step is crucial, otherwise the other remaining agents may not obtain enough jobs. It is obvious that if agent $a_i$ gets assigned a bag, then her true utility satisfies

$$u_i(X_i) = \sum_{j_l \in X_i} u_i(j_l) = u'_i(X_i) \geqslant \frac{\beta}{\beta+2} \cdot \gamma_i.$$

The major technical difficulty of our algorithm is to prove that everyone can obtain a bag.

---

**Algorithm 2.** BagFilling Procedure

---

**Input:** An FISP instance $\mathcal{I} = (J, A, \mathbf{u}_A)$ such that $u_i(j_k) \leqslant \frac{\beta}{\beta+2} \cdot \gamma_i$ for all $a_i \in A$ and $j_k \in J$; $\beta$-approximation algorithm for IS functions; Thresholds $(\gamma_1, \cdots, \gamma_m)$.

**Output:** $\frac{\beta}{\beta+2}$-MMS schedule $\mathbf{X} = (X_1, \cdots, X_m)$.
 1: Initialize $A' = A, J' = J$, and obtain approximate utility functions $u'_i$ for all $a_i \in A$.
 2: **while** $A' \neq \emptyset$ and $J' \neq \emptyset$ **do**
 3:   Set $B = \emptyset$.
 4:   **while** $u'_i(B) < \frac{\beta}{\beta+2} \cdot \gamma_i$ for all $a_i \in A'$ and $J' \neq \emptyset$ **do**
 5:     Let $j_k$ be an arbitrary job in $J'$. Set $B = B \cup \{j_k\}$ and $J' = J' \setminus \{j_k\}$.
 6:   **end while**
 7:   Let $a_i$ be an arbitrary agent such that $u'_i(B) \geqslant \frac{\beta}{\beta+2} \cdot \gamma_i$.
 8:   Let $X_i \subseteq B$ be a feasible subset such that $\sum_{j_l \in X_i} u_i(j_l) = u'_i(B)$.
 9:   Set $J' = J' \cup (B \setminus X_i)$ and $A' = A' \setminus \{a_i\}$.
10: **end while**

---

**Lemma 3.** *Setting $\gamma_i = \mathsf{MMS}_i$ for all $a_i \in A$, Algorithm 2 returns a $\frac{\beta}{\beta+2}$-MMS schedule.*

*Proof.* As we have discussed, it suffices to prove that at the beginning of any round of the outer while loop, there are sufficiently many remaining jobs in $J'$ for every remaining agent in $A'$, i.e.,

$$u'_i(J') \geqslant \frac{\beta}{\beta+2} \gamma_i, \text{ for any } a_i \in A'.$$

To prove the above inequality, in the following, we actually prove a stronger argument.

**Claim 1.** *For any $a_i \in A'$, let $\mathbf{X}' = (X'_1, \cdots, X'_m)$ be a feasible MMS schedule for $a_i$. Then there exists $k \in [m]$, such that $u_i(X'_k \cap J') \geqslant \frac{1}{\beta+2} \cdot \gamma_i$.*

Given Claim 1 and the $\beta$-approximation of $u'_i$, $u'_i(X'_k \cap J') \geqslant \frac{\beta}{\beta+2} \cdot \gamma_i$ and thus the lemma holds. We prove by contradiction and assume Claim 1 does not hold for agent $a_i$. Since $\mathbf{X}' = (X'_1, \cdots, X'_m)$ is a feasible MMS schedule for $a_i$, $u_i(X'_k) \geqslant \mathsf{MMS}_i = \gamma_i$ for all $k \in [m]$ and thus

$$\sum_{k \in [m]} u_i(X'_k) \geqslant m \cdot \gamma_i. \tag{1}$$

Denote by $(X_r)_{a_r \in A \setminus A'}$ the assignments that are allocated to $A \setminus A'$ in previous rounds by Algorithm 2, and for each $a_r$, let $j_{l_r}$ be the last item added to the bag $B$. Note that $j_{l_r} \in X_r$ otherwise $a_r$ will stop the inner while loop (Step 4) before $j_{l_r}$ was added. Moreover, since $a_i$ did not break the while loop either, $u_i'(X_r \setminus \{ j_{l_r} \}) < \frac{\beta}{\beta+2} \cdot \gamma_i$. Thus $u_i(X_r \setminus \{ j_{l_r} \}) \leqslant \frac{1}{\beta+2} \cdot \gamma_i$ as $u_i'$ is $\beta$-approximation of $u_i$. By the assumption that all jobs are small, i.e., $u_i(j_{l_r}) \leqslant \frac{\beta}{\beta+2} \cdot \gamma_i$, we have the following

$$u_i(X_r) = u_i(X_r \setminus \{ j_{l_r} \}) + u_i(j_{l_r}) < \frac{\beta+1}{\beta+2} \cdot \gamma_i. \tag{2}$$

If $u_i(X_k' \cap J') < \frac{1}{\beta+2} \cdot \gamma_i$ for all $k \in [m]$, then

$$
\begin{aligned}
\sum_{k \in [m]} u_i(X_k') &= \sum_{k \in [m]} \left( u_i(X_k' \cap J') + \sum_{a_r \in A \setminus A'} u_i(X_k' \cap X_r) \right) \\
&= \sum_{k \in [m]} u_i(X_k' \cap J') + \sum_{a_r \in A \setminus A'} \sum_{k \in [m]} u_i(X_k' \cap X_r) \\
&\leqslant \sum_{k \in [m]} u_i(X_k' \cap J') + \sum_{a_r \in A \setminus A'} u_i(X_r) \\
&< m \cdot \frac{1}{\beta+2} \cdot \gamma_i + (m - |A'|) \cdot \frac{\beta+1}{\beta+2} \cdot \gamma_i < m \cdot \gamma_i,
\end{aligned}
$$

where the first inequality is because the $X_k'$'s are disjoint and the second inequality is because of Equation (2). Thus we obtain a contradiction with Equation (1). □

### 3.1.3 Main Existential Theorem

Combining Lemma 2 and Lemma 3, it is not hard to prove the main existential result. In the full version Li *et al.* [2021], we will prove that our analysis is asymptotically tight.

---

**Algorithm 3.** Main Algorithm: Matching-BagFilling

---

**Input:** An arbitrary FISP instance $\mathcal{I} = (J, A, \mathbf{u}_A)$; $\beta$-approximation algorithm for IS functions; Thresholds $(\gamma_1, \cdots, \gamma_m)$.
**Output:** $\frac{\beta}{\beta+2}$-MMS schedule $\mathbf{X} = (X_1, \cdots, X_m)$.
1: Run Algorithm 1 on $\mathcal{I}$ with $(\gamma_1, \cdots, \gamma_m)$. Obtain $\mathcal{I}' = (J', A', \mathbf{u}_{A'})$ and $(X_r)_{a_r \in A \setminus A'}$.
2: Run Algorithm 2 on $\mathcal{I}'$ with $(\gamma_1, \cdots, \gamma_m)$ and the $\beta$-approximation algorithm. Obtain $(X_i)_{a_i \in A'}$.

---

**Theorem 1.** *Algorithm 3 with the optimal algorithm for IS functions (i.e., $\beta = 1$) and $\gamma_i = \mathsf{MMS}_i$ for all $a_i \in A$ returns a 1/3-MMS schedule for arbitrary FISP instance.*

Interestingly, in the independent and parallel work Hummel and Hetland [2021], via a similar bag-filling algorithm, the authors prove the existence of $1/3$-approximate MMS allocations under the context of graphically conflicting items. However, the two models in our work and theirs are not compatible in general.

### 3.2 Polynomial-time Implementation

Note that, in general, Algorithm 3 is not efficient, because if P $\neq$ NP, the computation of exact values for IS functions and MMS values cannot be done in polynomial time. For the special case when jobs are rigid or unit, IS functions can be computed in polynomial time. If the number of machines is constant, MMS values for rigid jobs can be computed in pseudo-polynomial time Chiarelli *et al.* [2020]. Thus, in this section, we deal with the general case. Of course, for IS functions, we can directly use the $\beta$-approximation algorithms, and the best-known approximation ratio is 0.644 Im *et al.* [2020]. Regarding the MMS barrier, instead of using their approximate values, we utilize a combinatorial trick similar with one used in Barman and Krishnamurthy [2020] such that without knowing their values, we can still execute our algorithm.

First, an important corollary of Lemma 2 and Lemma 3 is that if $\gamma_i \leqslant \mathsf{MMS}_i$ for some $a_i$, no matter what values are set for $\gamma_j, j \neq i$, Algorithm 3 always assigns a bag to $a_i$ such that $u_i(X_i) \geqslant \frac{\beta}{\beta+2} \gamma_i$.

**Lemma 4.** *For any $a_i$, if $\gamma_i \leqslant \mathsf{MMS}_i$, Algorithm 3 ensures that $u_i(X_i) \geqslant \frac{\beta}{\beta+2}\gamma_i$, regardless of $\gamma_{-i}$.*

We prove Lemma 4 in the full version Li *et al.* [2021]. Now, we are ready to introduce the trick. First, we set each $\gamma_i$ to be sufficiently large such that $\gamma_i \geqslant \mathsf{MMS}_i$ for all $a_i$. Then we run Algorithm 3. If we found some agent $a_i$ with $u_i(X_i) < \frac{\beta}{\beta+2}\gamma_i$, it means $\gamma_i$ is higher than $\mathsf{MMS}_i$ and we can decrease $\gamma_i$ by $0 < 1 - \epsilon < 1$ fraction and keep the other MMS values unchanged. We repeat the above procedure until everyone is satisfied $u_i(X_i) \geqslant \frac{\beta}{\beta+2}\gamma_i$. By Lemma 4, it must be that $\gamma_i \geqslant (1 - \epsilon)\mathsf{MMS}_i$ for all $a_i$. We summarize this in Algorithm 4, and it is straightforward to have the following theorem.

---

**Algorithm 4.** Efficient Implementation: Matching-BagFilling

---

**Input:** An arbitrary FISP instance $\mathcal{I} = (J, A, \mathbf{u}_A)$; $\beta$-approximation polynomial-time algorithm for IS functions; Thresholds $(\gamma_1 = \frac{u_1'(J)}{\beta}, \cdots, \gamma_m = \frac{u_m'(J)}{\beta})$; $0 < \epsilon < 1$.

**Output:** $\frac{\beta}{(\beta+2)}(1 - \epsilon)$-MMS schedule $\mathbf{X} = (X_1, \cdots, X_m)$.

1: Run Algorithm 3 on $\mathcal{I}$ with $(\gamma_1, \cdots, \gamma_m)$. Obtain $\mathbf{X} = (X_1, \cdots, X_m)$.
2: **while** there exist $a_i \in A$ such that $u_i'(X_i) < \frac{\beta}{\beta+2}\gamma_i$ **do**
3:     Set $\gamma_i = (1 - \epsilon)\gamma_i$.
4:     Run Algorithm 3 on $\mathcal{I}$ with $(\gamma_1, \cdots, \gamma_m)$ and update $\mathbf{X} = (X_1, \cdots, X_m)$.
5: **end while**

---

**Theorem 2.** *For any $0 < \epsilon < 1$, Algorithm 4 returns a $\frac{\beta}{\beta+2}(1 - \epsilon)$-MMS schedule for arbitrary FISP instance with an $\beta$-approximation algorithm for IS functions. The running time is polynomial with $|J|$, $|A|$ and $1/\epsilon$. Particularly, using the 0.64-approximation algorithm in Im et al. [2020], we have $0.24(1 - \epsilon)$-approximation polynomial-time algorithm.*

## 4 Approximately EF1 and PO Scheduling

### 4.1 Compatibility of EF1 and PO

In this section, we investigate the extent to which there is a schedule that is both EF1 and PO. We first show that EF1 and PO are not compatible even if jobs are rigid and valuations are unary, i.e., $u_i(j_k) = 1$ for all $a_i \in A$ and $j_k \in J$. That is no algorithm can return an EF1 and PO schedule for all instances. Fortunately, if the jobs have unit processing time, an EF1 and PO schedule exists and can be computed in polynomial time. This result continues to hold if the agents have weighted but identical utilities, i.e., $u_i(j_k) = u_r(j_k)$ for any job $j_k$ and any two agents $a_i$ and $a_r$. We sometimes ignore the subscript and use $u(\cdot)$ to denote the identical valuation.

---

**Algorithm 5.** $m$-Matching + Inner-Greedy

---

**Input:** An FISP instance $\mathcal{I} = (J, A, \mathbf{u}_A)$, where all jobs have unit processing time and all agents have identical valuation.

**Output:** EF1 and PO schedule $\mathbf{X} = (X_1, \cdots, X_m)$.

1: Construct graph $G(J \cup T, E)$, and compute a maximum weighted $m$-matching $\mathcal{M}^*$.
2: Define $J_t = \{ j \in J \mid (j, t) \in \mathcal{M}^* \}$ for each $t \in T$.
3: Set $X_1 = X_2 = \cdots = X_m = \emptyset$.
4: **for** $p = 1$ to $|T|$ **do**
5:     **if** $J_p \neq \emptyset$ **then**
6:         Sort $A$ in non-decreasing order of $u_i(X_i)$'s, and $J_p$ in non-increasing order of $u(j_k)$'s.
7:         **for** $i = 1$ to $|J_p|$ **do**
8:             Set $X_i = X_i \cup \{ j_i \}$.
9:         **end for**
10:     **end if**
11: **end for**

---

**Theorem 3.** *EF1 and PO are not compatible even if all jobs are rigid and all valuations are unary. If all jobs have unit processing time, Algorithm 5 returns an EF1 and PO schedule in polynomial time, as long as the valuations are identical.*

Here, we briefly discuss the intuition of Algorithm 5 in the following. At the heart of Algorithm 5, there are two tasks: (1) Find jobs with maximum weight to schedule in order to guarantee PO. (2) Assign these jobs to agents such that the assignment is EF1.

To solve the first task, we use bipartite matchings. Let $T_i = [r_i, d_i] = \{ r_i, r_i + 1, \cdots, d_i \}$ be the job interval for each $j_i \in J$, and $T = \bigcup_{1 \leqslant i \leqslant n} T_i$. We first construct a weighted bipartite graph $G(J \cup T, E)$, where each job $j_i \in J$ is a node on the left side, and a time slot $t_l \in T$ is a node on the right side. There is an edge $(j_i, t_l) \in E$ with weight $u(j_i)$ if and only if $t_l \in T_i$. For any set $\mathcal{B} \subseteq E$ and $v \in J \cup T$, let $\mathcal{B}(v) \subseteq E$ be the set of edges in $\mathcal{B}$ that intersects $v$. Next we define $m$-*matching* $\mathcal{M} \subseteq E$, which requires that $|\mathcal{M}(j_i)| \leqslant 1$ for all $j_i \in J$ and $|\mathcal{M}(t_l)| \leqslant m$ for all $t_l \in T$, where $m$ is the number of machines. $m$-matching is used to ensure that at each time slot at most $m$ jobs are processed. Accordingly, by computing a maximum weighted $m$-matching $\mathcal{M}^*$, we can find the set of jobs by scheduling which we can maximize the social welfare so that the resulting schedule is PO.

However, how shall we assign these jobs to agents? We partition these selected jobs into groups $\{J_1, \cdots, J_{|T|}\}$, and each group $J_t$ contains the jobs that are matched to time slot $t$ by $\mathcal{M}^*$. To maintain feasibility, each agent can get at most one job from each $J_t$. We process $\{J_1, \cdots, J_{|T|}\}$ one by one, and to satisfy EF1, the agent with low cumulative utility should obtain a better job in the next group. By induction, we can prove that eventually, the assignment is indeed EF1.

A final remark is about the running time. If the graph size is polynomial, by Bernhard and Vygen [2008], finding $\mathcal{M}^*$ can be done in polynomial time. However, $|T|$ can be exponentially large. Therefore, before running Algorithm 5, we first reduce an arbitrary instance to a *condensed* one by discarding some time slots which is essentially equivalent to the original one but only contains polynomial number of time slots. We defer this discussion completely to the full version Li *et al.* [2021].

## 4.2 Approximate EF1 and PO

Although EF1 and PO are only compatible in special cases, in this section we show that approximate EF1 and PO can be always satisfied. Theorem 4 is proved in the full version Li *et al.* [2021], where we introduce Nash social welfare and show that Nash social welfare maximizing schedule satisfies the desired properties.

**Theorem 4.** *For arbitrary FISP instance, there is a feasible schedule that is $1/4$-EF1 and PO. If all jobs have unit processing time, there is a feasible schedule that is $1/2$-EF1 and PO.*

## 5 Experiment

Finally, we evaluate the performance of Matching-BagFilling on randomly generated data sets, and compare it with the extensively adopted heuristic algorithm *Round-Robin*: Each agent takes turns to select a feasible unscheduled job that maximizes her marginal utility gain until no more jobs can be selected. Before conducting the experiments, we note that although Matching-BagFilling has good theoretical guarantee, for many concrete instances, it can be significantly improved. Since each agent only selects a bag with $1/3\text{MMS}_i$, there might be lots of unscheduled jobs that can be feasibly processed. Therefore, we first refine our algorithm as *Matching-BagFilling+*: (1) Run Matching-BagFilling; (2) Run Round-Robin on the remaining unscheduled jobs. We formally describe Round-Robin and Matching-BagFilling+ in the full version Li *et al.* [2021]. Note that Matching-BagFilling+ does not have better theoretical performance than Matching-BagFilling.

In our experiment, as shown in Figure 1, we randomly generate a set of rigid jobs $J$ ($|J| = 100, 500, 1000$) and a set of agents $A$ following some distribution, where for $i = 1, 2, 3, U/P/N.i$ means there are $|A| = 5 \times i$ agents whose values are randomly generated from a(n) Uniform, Poisson, or Normal distribution. For each setting, we generate 1000 instances. For each of them we run Matching-BagFilling+ (short for BAG+), Matching-BagFilling (short for BAG), and Round-Robin (short for RR) and record every agent $a_i$'s average utilities $u_i(\text{BAG+}), u_i(\text{BAG}), u_i(\text{RR})$. Then we compute the ratios $u_i(\text{BAG+})/u_i(\text{RR})$ and $u_i(\text{BAG})/u_i(\text{RR})$. Of course, if these ratios are greater than 1, it means our algorithms outperform the Round-Robin. We use two vertical line intervals to represent the range of all agents' ratios. It can be seen that in all experiments, Matching-BagFilling+ clearly outperforms Round-Robin, i.e., every agent gets higher utility in Matching-BagFilling+. Compared with the number of agents, if the number of jobs is small (e.g.

$|A| = 15$ and $|J| = 100, 500, 1000$), Matching-BagFilling has decent performance. However, as the number of jobs gets larger (e.g., $|A| = 5$ and $|J| = 100, 500, 1000$), Matching-BagFilling is clearly worse than Round-Robin, which is because too many jobs are left unscheduled. We defer a detailed description of our experiments and the discussion of results to the full version Li *et al.* [2021].

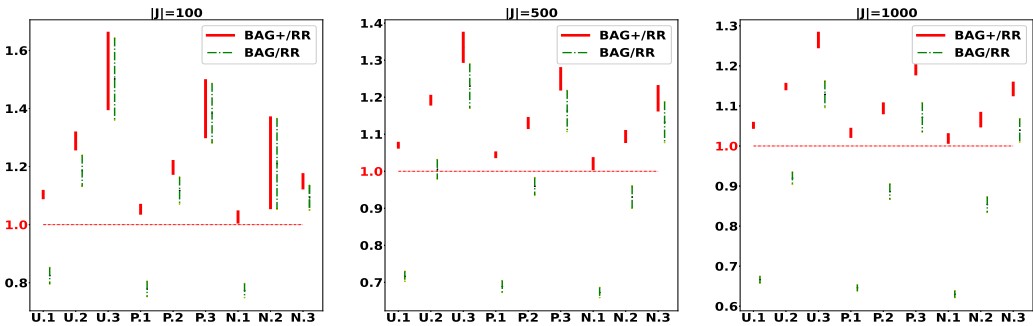

Figure 1: Experiments

## 6 Conclusion and Future Directions

In this work, we studied the fair scheduling problem for time-dependent resources, and designed constant approximation algorithms for MMS, EF1&PO and EF1&IO schedules. There are many open problems and future directions. An immediate direction is to improve our approximation ratios and investigate the limit of approximation algorithms for different settings. It is also interesting to impose other efficiency criteria on EF1 schedules, such as computing an EF1 schedule that maximizes social welfare. In this work, we have assumed the jobs are resources that bring utility to agents, and leave the case when jobs are chores for future study. Finally, it is of both theoretical interest and practical importance to consider the online setting when jobs arrive dynamically and the strategic setting when agents' valuations are private information.

## Acknowledgements and Funding

The authors thanks Warut Suksompong for reading a draft of this paper and for helpful discussions. Bo Li was partially funded by The Hong Kong Polytechnic University under Grant No. P0034420. Minming Li was partially supported by NSFC under Grant No. 11771365, and by Project No. CityU 11200518 from Research Grants Council of HKSAR.

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
