## Appendix

The appendix is organized as follows:

- Appendix A: In this section, we prove IS functions are XOS.
- Appendix B: In this section, we formally prove our main results in Section 3.
- Appendix C: In this section, we formally prove our main results in Section 4.
- Appendix D: In this section, we introduce a milder efficiency requirement IO and study its compatibility with EF1.
- Appendix E: In this section, we discuss our experiments in more details.

As we will discuss the approximation algorithms and the existences of EF1/PO/IO schedules in different settings, we introduce the following notations to simplify the description of different settings.

Regarding agents' utilities, FISP contains three cases, from the most special to the most general:

- Unweighted: $u_i(j_k) = 1$ for all $a_i \in A, j_k \in J$, i.e., agents have unary utility for jobs.
- Identical: $u_i(j_k) = u_r(j_k)$ for all $a_i, a_r \in A, j_k \in J$, i.e., all agents have the same utility for the same job.
- Non-identical: $u_i(j_k) \geqslant 0$ without any restrictions.

Regarding jobs, there are three cases:

- Unit: $p_i = 1$, for all $j_i \in J$, i.e., all jobs have unit processing time.
- Rigid: $r_i + p_i - 1 = d_i$, for all $j_i \in J$, i.e., the jobs need to occupy the entire time intervals between their release times and deadlines.
- Flexible: $r_i + p_i - 1 \leqslant d_i$, for all $j_i \in J$.

Note that unit jobs may not be rigid and rigid jobs may not be unit either. In the remainder of the appendix, we use notation FISP with ⟨utility type, job type⟩ to denote a certain case of the general FISP, e.g., FISP with ⟨unweighted, unit⟩ represents the case where the processing time of each job is 1 and each agent has unweighted utility function.

## A   Missing Materials in Section 2

A set function $f : 2^V \to \mathbb{R}$ defined on $V$ is called *fractionally subadditive* (XOS) if there is a finite set of additive functions $\{ f_1, \cdots, f_w \}$ such that $f(S) = \max_{1 \leqslant i \leqslant m} f_i(S)$ for any $S \subseteq V$.

**Lemma 5.** *IS functions are XOS.*

*Proof.* Let $u$ be an IS function defined on job set $J = \{ j_1, \cdots, j_n \}$ with individual utility $(v_1 = u(j_1), \cdots, v_n = u(j_n))$. To show $u$ is XOS, it suffices to define a finite set of additive functions on $J$. For each feasible job set $T \subseteq J$, define additive function $f_T$ such that $f_T(j_i) = v_i$ if $j_i \in T$ and $f_T(j_i) = 0$ otherwise. Therefore, for any $S \subseteq T$,

$$u(S) = \max_{T \subseteq S:T \text{ is feasible}} \sum_{j_i \in T} v_i = \max_{T \subseteq S:T \text{ is feasible}} f_T(T) = \max_{T \subseteq S:T \text{ is feasible}} f_T(S) = \max_{T \text{ is feasible}} f_T(S),$$

where the last equality is because any subset of a feasible job set is also feasible. Thus $u$ is XOS. □

## B   Missing Materials for MMS Scheduling in Section 3

### B.1   Proof of Lemma 1

*Proof.* Let $\mathcal{I} = (J, A, \mathbf{u}_A)$ be an arbitrary instance of FISP with $J = \{ j_1, \cdots, j_m \}$ and $|A| = m$. To show that $\mathsf{MMS}_i(J \setminus \{ j_k \}, m - 1) \geqslant \mathsf{MMS}_i(J, m)$ holds for any $j_k \in J, a_i \in A$, we consider an arbitrary agent $a_i$. Let $\mathbf{X} = (X_1, X_2, \cdots, X_m)$ be a feasible schedule for $a_i$, i.e.,

$\min_{X_r \in \mathbf{X}} u_i(X_r) = \mathsf{MMS}_i$. Consider an arbitrary job $j_k$, assume that $j_k \in X_l$. Then remove job set $X_l$ from $\mathsf{MMS}_i$ schedule. This generates a new schedule, denoted by $\mathbf{X}' = \{ X_1', X_2', \cdots, X_{m-1}' \}$. It is easy to see that $\mathbf{X}'$ is a feasible schedule to the instance with $m-1$ agents and the job set $J \setminus \{ J_k \}$. This implies that $\mathsf{MMS}_i(J \setminus \{ j_k \}, m-1) \geqslant \min_{X_r' \in \mathbf{X}'} u_i(X_r')$. Note that $\min_{X_r' \in \mathbf{X}'} u_i(X_r') \geqslant \mathsf{MMS}_i(J, m)$. Therefore, we have

$$\mathsf{MMS}_i(J \setminus \{ j_k \}, m-1) \geqslant \min_{X_r' \in \mathbf{X}'} u_i(X_r') \geqslant \mathsf{MMS}_i(J, m).$$

In the case where $j_k \notin \bigcup_{r \in [m]} X_r$, we remove an arbitrary job set from $\mathbf{X}$ and the above analysis still works. $\qquad\square$

## B.2 An Example for Matching-BagFilling and Matching-BagFilling+

It is obvious that our analysis is tight when all jobs have tiny values such that whenever an agent obtains a bag her value is exactly or slightly greater than $\frac{1}{3}\mathsf{MMS}_i$. In the following, we present an instance such that even without the preprocessing procedure and the last agent takes away all remaining jobs, everyone obtains exactly $\frac{1}{3}\mathsf{MMS}_i + \epsilon$. Accordingly, the instance proves that "Matching-BagFilling+ does not have better theoretical performance than Matching-BagFilling" as claimed in Section 5.

Consider the following instance with $|A| = m$ agents where $m$ is a sufficiently large even number.

The job set $J$ can be classified into the following categories:

- $J_1 = \{ j_1^1, j_2^1, \cdots, j_m^1 \}$: There are $m$ rigid jobs in $J_1$. Every job in $J_1$ has the same job interval $[1, 2]$. For every job in $J_1$, $a_m$ has the same utility gain $\frac{1}{3} + \frac{1}{m}$. For every job in $J_1$, all agents in $A \setminus \{ a_m \}$ have the same utility gain $\frac{2}{3} + \frac{1}{m}$;

- $J_2 = \{ j_1^2, j_2^2, \cdots, j_{m-1}^2 \}$: There are $m-1$ rigid jobs in $J_2$. Every job in $J_2$ has the same job interval $[3, \frac{m}{2} + 2]$. For every job in $J_2$, all agents in $A$ have the same utility gain $\frac{1}{3}$;

- $J_3 = \{ j_1^3, j_2^3, \cdots, j_{m-1}^3 \}$: There are $m$ unit jobs in $J_3$. Every job in $J_3$ has the same job interval $[3, m + 2]$. For every job in $J_3$, all agents in $A$ have the same utility gain $\frac{1}{3m}$;

- $J_4 = \bigcup_{r \in [m]} J_4^r$: There are $m$ group rigid jobs in $J_4$. Each group $J_4^r, r \in [m]$, contains $m$ rigid jobs. Assume that $J_4^r = \{ j_{r1}^4, j_{r2}^4, \cdots, j_{rm}^4 \}, \forall r \in [m-1]$. A job $j_{ri}^4 \in J_4^r, i \in [m]$ has the job interval $[m + 3 + i, m + 4 + i]$. Assume that $J_4^m = \{ j_{m1}^4, j_{m2}^4, \cdots, j_{mm}^4 \}$. A job $j_{mi}^4 \in J_3^m$ has the job interval $[m + 4 + i, m + 5 + i]$. In total, there are $m^2$ jobs in $J_4$. For every job in $J_4$, $a_m$ has the same utility gain $\frac{1}{3m}$. For every job in $J_4$, all agents in $A \setminus \{ a_m \}$ have the same utility gain $0$.

Let us focus on $a_m$ first. The upper bound of $\mathsf{MMS}_m$ is:

$$\frac{1}{m} \cdot \left( (\frac{1}{3} + \frac{1}{m}) \cdot m + \frac{m-1}{3} + \frac{1}{3m} \cdot m + \frac{1}{3m} \cdot m^2 \right) = 1 + \frac{1}{m}.$$

We consider the schedule $\mathbf{X} = (X_1, \cdots, X_m)$, where $X_i = \{ j_i^1, j_i^2 \} \cup J_4^i, \forall i \in [m-1]$ and $X_m = \{ j_m^1 \} \cup J_3 \cup J_4^m$ (See Figure 2). It is not hard to see that $\mathbf{X}$ is a feasible schedule and $\min_{i \in [m]} u_m(X_i) = u_m(X_m) = 1 + \frac{1}{m}$. Therefore, $\mathbf{X}$ is a feasible schedule that obtains the value $1 + \frac{1}{m}$ which is also the upper bound of $\mathsf{MMS}_m$. Thus, $\mathsf{MMS}_m = 1 + \frac{1}{m}$. Hence, once $a_m$ values the bag greater than or equal to $\frac{1}{3} + \frac{1}{3m}$, $a_m$ will take the bag away.

Now, we consider an arbitrary agent $a_i \in A \setminus \{ a_m \}$. Since all agents in $A \setminus \{ a_m \}$ have utility gain $0$ for all jobs in $J_4$, we can ignore the job set $J_4$. Therefore, the upper bound of $\mathsf{MMS}_i, \forall i \in [m-1]$ is:

$$\frac{1}{m} \left( (\frac{2}{3} + \frac{1}{m}) \cdot m + (\frac{m-1}{3}) + (\frac{1}{3m}) \cdot m \right) = 1 + \frac{1}{m}.$$

We consider the schedule $\mathbf{X}' = (X_1', \cdots, X_m')$, where $X_i' = \{ j_i^1, j_i^2 \}, \forall i \in [m-1]$ and $X_m' = \{ j_m^1 \} \cup J_3$. It is not hard to see that $\mathbf{X}'$ is a feasible schedule and $u_i(X_k') = u_i(X_r') = 1 + \frac{1}{m}, \forall k, r \in [m], \forall i \in [m-1]$. Therefore, $\mathbf{X}'$ is a feasible schedule that obtains the value $1 + \frac{1}{m}$ which is also an upper bound of $\mathsf{MMS}_i, \forall i \in [m-1]$. Thus, $\mathsf{MMS}_i = 1 + \frac{1}{m}, \forall i \in [m-1]$. Hence, once agent $a_i, \forall i \in [m-1]$, values the bag greater than or equal to $\frac{1}{3} + \frac{1}{3m}$, $a_i$ will take the bag away.

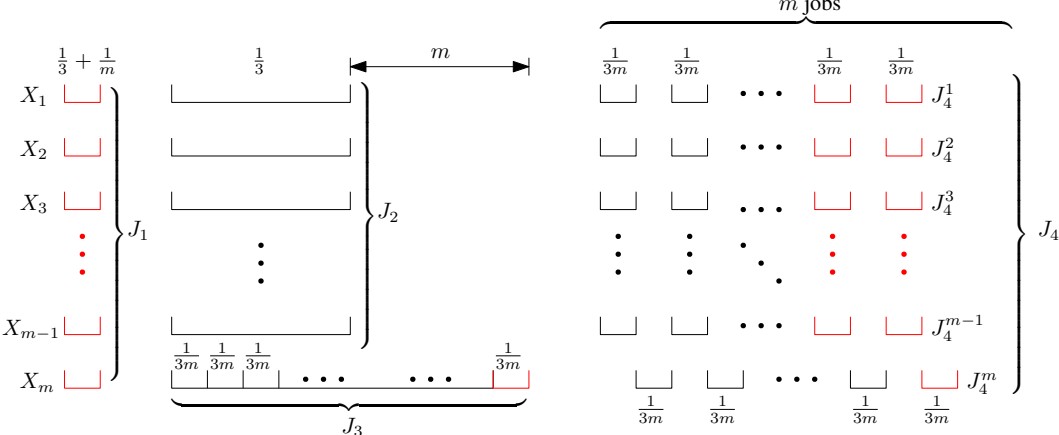

Figure 2: Illustration for the tight instance for Algorithm 3. The above schedule is **X** which is also the MMS schedule for agent $a_m$. The red jobs are the remaining jobs at the end of the $(m-1)$-th round of Algorithm 3 with the specified job sequence described in the "The specified job sequence" paragraph.

**The specified job sequence**   Now, we consider the following job sequence. In the first round, Algorithm 3 adds $J_1^4 \setminus \{j_{1(m-1)}^4, j_{1m}^4\}$ to the bag, and then adds $j_1^3, j_{m1}^4$ to the bag, and then adds $j_1^2$ to the bag, i.e., $\mathsf{BAG} = \{j_{11}^4, j_{12}^4, \cdots, j_{1(m-2)}^4\} \cup \{j_1^3, j_{m1}^4\} \cup \{j_1^2\}$. It is not hard to see that BAG is a feasible job set and all agents in $A \setminus \{a_m\}$ value the bag exactly $\frac{1}{3} + \frac{1}{3m}$. Without loss of generality, we assume that $a_1$ takes the bag away at the end of the first round. In the $l$-th round, $2 \leqslant l \leqslant m-1$, Algorithm 3 first adds $J_l^4 \setminus \{j_{l(m-1)}^4, j_{lm}^4\}$, and then adds $j_l^3, j_{ml}^4$, and then adds $j_l^2$ to the bag. Without loss of generality, we assume that $a_l$ takes the bag away at the end of the $l$-th round, where $2 \leqslant l \leqslant m-1$. Note that, at the end of the $(m-1)$-th round, all agents in $A \setminus \{a_m\}$ obtain the utility gain exactly $\frac{1}{3} + \frac{1}{3m}$.

It is not hard to see that, at the end of the $(m-1)$-th round,

$$J' = J_1 \cup \{j_m^3\} \cup \{j_{mm}^4\} \cup \{j_{1(m-1)}^4, j_{1m}^4\} \cup \{j_{2(m-1)}^4, j_{2m}^4\} \cup \cdots \{j_{(m-1)(m-1)}^4, j_{(m-1)m}^4\}.$$

See the red jobs in Figure 2. Thus, $u_m(J') = (\frac{1}{3} + \frac{1}{m}) + \frac{1}{3m} + \frac{3}{3m} = \frac{1}{3} + \frac{7}{3m}$.

Therefore, everyone obtains exactly $\frac{1}{3}\mathsf{MMS}_i + \epsilon$ at the end of Algorithm 3. Moreover, it is not hard to see that if we run round-robin procedure at the end of Algorithm 3, the utility gains of all agents in $A \setminus \{a_m\}$ will be increased but the utility gain of $a_m$ is not able to be further improved. Thus, the above instance implies that "Matching-BagFilling+ does not have better theoretical performance than Matching-BagFilling".

### B.3   Proof of Lemma 4

Note that the algorithm only ensures that agent $a_i$ with $\gamma_i \leqslant \mathsf{MMS}_i$ can obtain a bag but not everyone. This is natural as if for some $a_j \neq a_i$ and $\gamma_j$ is super large compared with $\mathsf{MMS}_j$, $a_j$ will never stop the algorithm and get a bag.

Recall that we can assume that there is no large job in the instance, i.e., $u_i(j_k) \leqslant \frac{\beta}{\beta+2} \cdot \gamma_i$, where $0 \leqslant \beta \leqslant 1$. Observe that if agent $a_i$ gets assigned a bag, then her true utility satisfies:

$$u_i(X_i) = \sum_{j_l \in X_i} u_i(j_l) = u_i'(X_i) \geqslant \frac{\beta}{\beta+2}\gamma_i.$$

The above inequality also holds no matter whether $\gamma_i \leqslant \mathsf{MMS}_i$ or not. Similar as the proof of Lemma 3, the core is to prove that $a_i$ can be guaranteed to obtain a bag as long as $\gamma_i \leqslant \mathsf{MMS}_i$. We consider the $R$-th round of the outer while loop of Algorithm 4 (line 2-5) in which the value of $\gamma_i$ is decreased below $\mathsf{MMS}_i$. In the $R$-th round of Algorithm 4 (line 2-5), we assume that the order of the agents that break the while loop of Algorithm 2 (line 4-10) is $\{a_1, \cdots, a_{i-1}, a_i, \cdots\}$. It suffices to

prove that at the beginning of the $i$-th while loop of Algorithm 2 (line 4-10), there are sufficiently many remaining jobs in $J'$ for the agent $a_i$, i.e.,

$$u_i'(J') \geqslant \frac{\beta}{\beta+2} \cdot \gamma_i, \forall a_i \in A'.$$

Similar as the proof of Lemma 3, we prove the following stronger claim. Given the following claim and the $\beta$-approximation of $u_i'$, we have $u_i'(X_k' \cap J') \geqslant \frac{\beta}{\beta+2} \cdot \gamma_i$ Therefore Lemma 4 holds.

**Claim 2.** *For any $a_i \in A'$ with $\gamma_i \leqslant \mathsf{MMS}_i$, let $\mathbf{X}' = \{X_1', \cdots, X_m'\}$ be a feasible MMS schedule for $a_i$. Then, there exists $k \in [m]$ such that $u_i(X_k' \cap J') \geqslant \frac{1}{\beta+2} \cdot \gamma_i$, where $\gamma_i \leqslant \mathsf{MMS}_i$.*

*Proof.* We consider an arbitrary agent $a_i$. Since $\mathbf{X}' = (X_1', X_2', \cdots, X_m')$ is a feasible MMS schedule for $a_i$, we have $u_i(X_k') \geqslant \mathsf{MMS}_i \geqslant \gamma_i, \forall k \in [m]$ and therefore

$$\sum_{k=1}^{m} u_i(X_k') \geqslant m \cdot \mathsf{MMS}_i \geqslant m \cdot \gamma_i. \tag{3}$$

Same as the proof of Lemma 3, the key idea of the proof is to show that agent $a_i$ values the bundles that are taken by the agents before $a_i$ less than $\frac{\beta+1}{\beta+2} \cdot \gamma_i$, i.e.,

$$u_i(X_r) < \frac{\beta+1}{\beta+2} \cdot \gamma_i, \forall r \in [i-1]. \tag{4}$$

We consider an arbitrary bundle that is taken by agent $a_r, r \in [i-1]$ and assume that job $j_r$ is the last job added to the Bag. Since $a_i$ did not break the while loop, we have $u_i'(X_r \setminus \{j_r\}) < \frac{\beta}{\beta+2} \cdot \gamma_i$. This implies that $u_i(X_r \setminus \{j_r\}) \leqslant \frac{1}{\beta+2} \cdot \gamma_i$. Since all jobs are small, i.e., $u_i(j_r) \leqslant \frac{\beta}{\beta+2} \cdot \gamma_i$, we have

$$u_i(X_r) = u_i(X_r \setminus \{j_r\}) + u_i(j_r) < \frac{\beta+1}{\beta+2} \cdot \gamma_i.$$

Therefore, Equation (4) holds. To help understand the following proof, an example is shown in Figure 3. Every rectangle in Figure 3 represents a job in $J$. The area of every rectangle $j_l$ in Figure 3 represents the value of $u_i(j_l)$. The non-white rectangles represent the jobs that are assigned to some agents in $\{a_1, \cdots, a_{i-1}\}$. According to Equation (4), the total area of non-white rectangles in Figure 3 is at most $\frac{(\beta+1)(i-1)}{\beta+2}\gamma_i$, i.e., $\sum_{r=1}^{i-1} u_i(X_r) < \frac{(\beta+1)(i-1)}{\beta+2}\gamma_i$. According to Equation (3), the total area of rectangles in Figure 3 is at least $m\gamma_i$. Therefore, the total area of white rectangles in $\{X_1', \cdots, X_m'\}$ is at least $m\gamma_i - \frac{(\beta+1)(i-1)}{\beta+2}\gamma_i$, i.e.,

$$\sum_{r=1}^{m} u_i(X_r' \setminus \bigcup_{l \in [i-1]} X_l) > m\gamma_i - \frac{(\beta+1)(i-1)}{\beta+2}\gamma_i \geqslant \frac{m+\beta+1}{\beta+2}\gamma_i, \tag{5}$$

where the last inequality is due to $i \leqslant m$.

According to Equation (5), the total area of white rectangles is at least $\frac{m+\beta+1}{\beta+2}\gamma_i$. There must exist an $r \in [m]$ such that $u_i(X_r' \cap J') \geqslant \frac{m+\beta+1}{m(\beta+2)}\gamma_i$. Therefore, Claim 2 holds. $\square$

## C  Missing Materials for EF1 and PO Scheduling in Section 4

### C.1  The Impossible Result for Theorem 3

In this subsection, we prove the first part of Theorem 3 (See Lemma 6).

**Lemma 6.** *EF1 and PO are not compatible for FISP with $\langle$unweighted, rigid$\rangle$, i.e., no algorithm can return a feasible schedule that is simultaneously EF1 and PO for all FISP with $\langle$unweighted, rigid$\rangle$ instances.*

*Proof.* To prove Lemma 6, we show that any PO schedule must not be an EF1 schedule for the instance in Figure 4. We consider an arbitrary PO schedule, denoted by $\mathbf{X} = (X_1, \cdots, X_m)$ and let $X_0 = J \setminus \bigcup_{i \in [m]} X_i$. We claim that $\mathbf{X}$ must satisfy the following two properties:

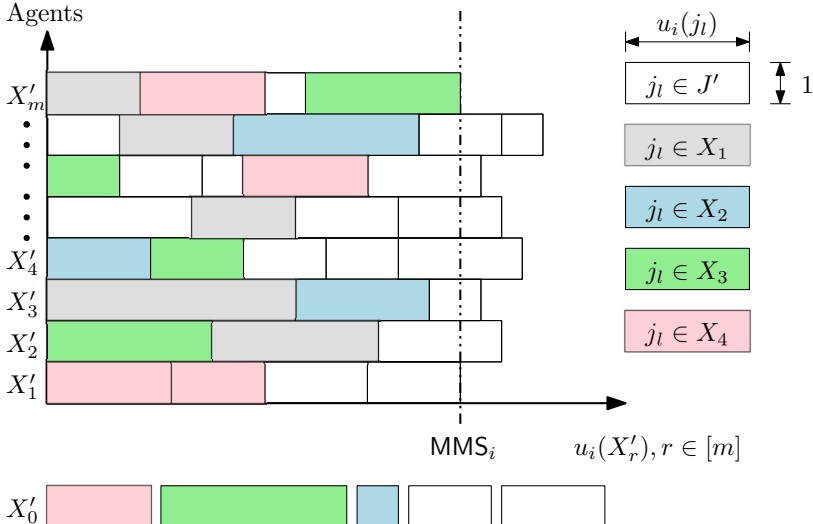

Figure 3: Illustration of Claim 2. The schedule is the feasible schedule $\mathbf{X}'$ which implies that job set $X'_r$ is a feasible for all $r \in [m]$. Every rectangle represents a job. The width of rectangle $j_l$ is the value of $u_i(j_l)$ while the height is 1. The area of rectangle $j_l$ is also the value of $u_i(j_l)$. The four agents $a_1, a_2, a_3, a_4 \in \{a_1, \cdots, a_{i-1}\}$. The non-white rectangles represent the jobs that are assigned in some agents in $\{a_1, \cdots, a_{i-1}\}$ in schedule $\mathbf{X}$, e.g., the gray, blue, green, pink rectangles are the jobs that are assigned to $a_1, a_2, a_3, a_4$, respectively. Recall that $\mathbf{X}$ is the schedule returned by Algorithm 2. The white rectangles are the jobs in $J'$. In Claim 2, we show that there exist a $r \in [m]$ such that total area of white rectangles in $X'_r$ is at least $\frac{1}{\beta+2}\gamma_i$.

1. $\exists i \in [m]$ such that $X_i = J_1$;

2. $X_0 = \emptyset$.

We first prove that there exists an $i \in [m]$ such that $X_i = J_1$. Suppose, towards to the contradiction, that there is no $i \in [m]$ such that $X_i = J_1$. Note that there must exist $i \in [m]$ such that $X_i \cap J_1 \neq \emptyset$ otherwise $\mathbf{X}$ is not a PO schedule. Now we consider the job set $X_l$ such that $X_l \cap J_1 \neq \emptyset$. In the case where no job set except $X_l$ in $\mathbf{X}$ contains jobs in $J_1$, we can construct another feasible schedule $\mathbf{X}' = \mathbf{X} \cup \{J_1\} \setminus X_l$. It is easy to see that $u_i(X'_i) \geqslant u_i(X_i)$ for all $a_i \in A$ and $u_l(J_1) > u_l(X_l)$ for agent $a_l$. This implies that $\mathbf{X}$ is not a PO schedule. In the case where there exist another one or two subsets $X_r, X_p \in \mathbf{X}$ such that $X_r, X_p \cap J_1 \neq \emptyset$. Since there are only three jobs in $J_1$, there are at most three job sets in $\mathbf{X}$ that contains some job in $J_1$. Without loss of generality, we assume that both $X_r$ and $X_p$ exist. Since every job in $J_2$ overlaps with every job in $J_1$, we have $X_l, X_r, X_p \cap J_2 = \emptyset$. Therefore, $|X_l| = |X_r| = |X_p| = 1$. Since there are $m - 1$ long jobs and every job set in $\mathbf{X} \setminus (X_l \cup X_r \cup X_p)$ contains only one job in $J_2$, $X_0$ contains two jobs from $J_2$. Now we can construct another feasible schedule $\mathbf{X}' = (X'_1, \cdots, X'_m)$ in following way: move all jobs in $X_r \cup X_p$ to $X_l$; assign one of two jobs in $X_0$ to $X_r$ and another one to $X_p$; keep the remaining job sets same as the corresponding one in $\mathbf{X}$. It is easy to see that $u_i(X'_i) \geqslant u_i(X_i)$ for all $a_i \in A$ and $u_l(X'_l) = u_l(J_1) > u_l(X_l)$. This implies that $\mathbf{X}$ is not a PO schedule.

Therefore, we can assume that there must exist an $i \in [m]$ such that $X_i = J_1$. Without loss of generality, we assume that $X_1 = J_1$. Now we show that the second property holds. Since $|J_2| = m - 1$, $X_0 \neq \emptyset$ implies that there must exist a job set $X_l \in \mathbf{X}$ such that $X_l = \emptyset$. This would imply that $\mathbf{X}$ is not a PO schedule. Since $\mathbf{X}$ holds the above two properties, we assume that every remaining agent in $A \setminus \{a_1\}$ will receive exactly one job in $J_2$. Without loss of generality, we assume that $X_i = \{j_{i+2}\}$. Therefore, we have $\mathbf{X} = (X_1, \cdots, X_m)$, where $X_1 = J_1, X_2 = \{j_4\}, \cdots, X_m = \{j_{m+2}\}$. Since $u_i(X_1 \setminus \{j\}) = 2 > u_i(X_i) = 1, \forall a_i \in A \setminus \{a_1\}, \forall j \in X_1$, $\mathbf{X}$ is not an EF1 schedule. $\qquad\square$

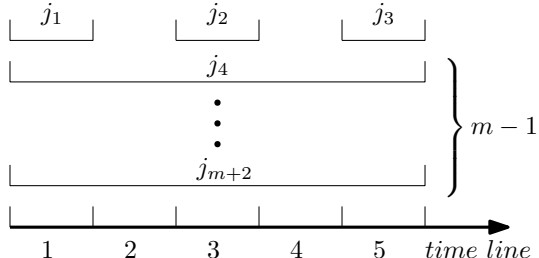

Figure 4: Instance for Lemma 6. There are $|A| = m$ agents and $|J| = m + 2$ jobs with $m \geqslant 2$. Job set $J$ can be partitioned as $J_1 \cup J_2$, where $J_1 = \{j_1, j_2, j_3\}$ and $J_2 = \{j_4, j_5, \cdots, j_{m+2}\}$. Each job $J_1$ has unit processing time and each job $J_2$ has processing time 5. All jobs are rigid such that $j_i \in J_1$ needs to occupy the entire time slot $2i - 1$, where $i \in \{1, 2, 3\}$. And $j \in J_2$ occupies the entire time period from 1 to 5.

## C.2 The Algorithm for Theorem 3

In this subsection, we mainly show Theorem 5, which is the second part of Theorem 3. Before give the proof of Theorem 5, we first give the definition of the *condensed instance* which is used to improve the running time.

Given an arbitrary instance of FISP with ⟨identical, unit⟩, denoted by $I$, for each job $j_i \in J$, let $\mathcal{T}_i$ be the set of time slots included in the job interval of $j_i$, i.e., $\mathcal{T}_i = \{r_i, r_i + 1, \cdots, d_i\}$. Let $\mathcal{T}$ be the set of *condensed time slots* (Definition 4). We construct another instance, denoted by $I'$, by condensing $T_i$, i.e., for every job in $J$, $T_i = \mathcal{T}_i \cap \mathcal{T}$. We show that these two instances are equivalent (Lemma 7). Let $J'$ be the set of jobs in the instance $I'$.

**Definition 4.** *Let $T$ be the condensed time slots set.*
$$T = \bigcup_{1 \leqslant l \leqslant n} \{d_l - n + 1, d_l - n + 2, \cdots, d_l\}$$
*where $d_l$ is the deadline of job $j_l$.*

To prove Lemma 8, it suffices to prove the following lemma.

**Lemma 7.** *Let $\hat{J} \subseteq J$ be an arbitrary subset of jobs in the instance $I$. Let $\hat{J}' \subseteq J'$ be the corresponding jobs in the instance $I'$. Then, $\hat{J}$ is a feasible job set if and only if $\hat{J}'$ is a feasible job set.*

*Proof.* ($\Leftarrow$) This direction is straightforward.

($\Rightarrow$) To prove this direction, we define a *job block* as a maximal set of consecutive jobs such that they are scheduled after each other. Since $\hat{J}$ is a feasible job set, there is a feasible schedule for all jobs in $\hat{J}$. We start from the first job $j_l \in \hat{J}$ which is scheduled in time slot $t_l$ such that $t_l \notin T$, we show that we can always shift this job block to the right. Time slot $t_l \notin T$ implies that $t_r$ is in a distance more that $n$ from any elements in deadline set $D = \bigcup_{j_q \in J} \{d_q\}$. We can shift the job block $j_l$ to the right. An example is shown in Figure 5. We show that we can always shift $j_l$ to the right until:

- Either job $j_l$ is scheduled in a time slot in $T$.

- Or the job block starting from $j_l$ reaches another scheduled job and form a bigger job block.

If job $j_l$ is scheduled in a time slot in $T$, then the lemma follows. If the job block starting from $j_l$ reaches another scheduled job and form a bigger job block, we keep shifting the bigger job block to the right unit $j_l$ is scheduled in a time slot $t'_l \in T$. Note that no job would miss its deadline, since the distance between $t'_l$ and any deadline in $D$ exceeds $n$ which implies that there is enough time slots to schedule all jobs in the current job block. □

**Lemma 8.** *For an arbitrary instance of FISP with ⟨identical, unit⟩, if there is a polynomial-time algorithm that returns an EF1 and PO schedule for all condensed instances, there also exists one for non-condensed instances.*

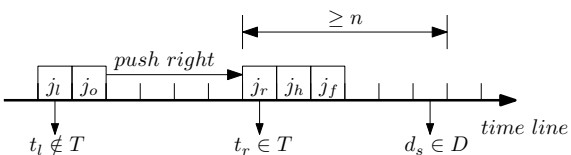

Figure 5: Illustration of Lemma 7. Initially, job $j_l$ is scheduled in the time slot $t_r$ which is not in $T$. The job block starting from $j_l$ only includes two jobs: $j_l$ and $j_o$. We can always shift job $j_l$ to the right until $j_l$ is scheduled in a time slot in $T$. Or the leftmost time slot in $T$ is occupied by a certain job $j_r$. The job block starting from $j_r$ contains three jobs: $j_r$, $j_h$ and $j_f$. We can still shift the merged job block, which contains $j_l, j_o, j_r, j_h, j_f$, to the right, since the distance between time slot $t_r$ and any deadline in $D$ exceeds $n$.

**Theorem 5.** *Given an arbitrary instance of FISP with $\langle$identical, unit$\rangle$, Algorithm 5 returns a schedule that is simultaneously EF1 and PO in polynomial time.*

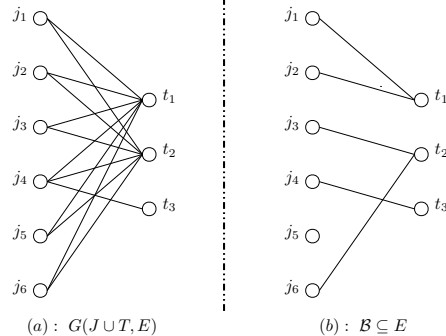

Figure 6: An example of the bipartite graph $G(J \cup T, E)$ and a corresponding maximum $m$-matching, where $J = \{j_1, \cdots, j_6\}$, $A = \{a_1, a_2\}$, $T = \{t_1, t_2, t_3\}$. Assume all jobs have identical utility to the agents. The job intervals are $T_1 = T_2 = T_3 = T_5 = T_6 = \{t_1, t_2\}$, and $T_4 = \{t_1, t_2, t_3\}$. A possible maximum weighted $m$-matching $\mathcal{M}^*$ is shown one the right, according to which the jobs are partitioned as $J_1 = \{j_1, j_2\}$, $J_2 = \{j_3, j_6\}$, $J_3 = \{j_4\}$. Then, $X_0 = J \setminus (J_1 \cup J_2 \cup J_3) = \{j_5\}$.

Arbitrarily fix a maximum weighted $m$-matching $\mathcal{M}^*$. For any $t \in T$, let $J_t$ be the set of jobs which are matched with time slot $t$, i.e., $J_t = \{j \in J \mid (j,t) \in \mathcal{M}^*\}$. Note that the $J_t$'s are mutually disjoint. Therefore we can refer $t$ as the *type* of jobs in $J_t$. An example can be found in Figure 6 (a). The key idea of Algorithm 5 is to first use the above procedure to find the job set with the maximum total weight that can be processed and classify jobs into different types, and then greedily assign each type of jobs to the agents. The jobs that are not matched by $\mathcal{M}^*$, i.e., $J \setminus (\bigcup_{t \in T} J_t)$, are kept unallocated and will be assigned to charity.

Let $\mathbf{X} = (X_1, \cdots, X_m)$ be the schedule returned by Algorithm 5 and let $X_0 = J \setminus \bigcup_{i \in [m]} X_i$. Note that although the agents have identical utilities, we sometimes use $u_i$ for agent $a_i \in A$ to make the comparison clear.

**Lemma 9.** *For any $a_i, a_k \in A$, $u_i(X_i) \geqslant u_i(X_k \setminus \{j_l\})$ for some $j_l \in X_k$.*

*Proof.* We prove the lemma by induction. Let $X_i^p$ be the set of jobs assigned to agent $a_i$ after the $p$-th round of Algorithm 5, and $\mathbf{X}^p = (X_1^p, \cdots, X_m^p)$.

*Base Case.* When $p = 1$, each agent gets at most one job as $|J_t| \leqslant m$ for all $t \in T$, and thus $\mathbf{X}^1$ is EF1.

*Induction Hypothesis.* For any $p > 1$, after the $p$-th round of Algorithm 5, suppose Lemma 9 holds, i.e., for any $a_i, a_k \in A$, there exists a job, denoted by $j_h$, in $X_i^p$ such that $u_k(X_k^p) \geqslant u_k(X_i^p \setminus \{j_h\})$.

Now we consider the $(p+1)$-th round. Arbitrarily fix two agents $a_i, a_k \in A$ and without loss of generality assume $u_i(X_i^p) \geqslant u_k(X_k^p)$. In the following we prove that after this round, $a_i$ and $a_k$ continue not to envy each other for more than one item. Note that, in the $(p+1)$-th round, $a_k$ chooses a job from $J_{p+1}$ before $a_i$.

Suppose that $j_{\hat{k}}$ is assigned to $a_k$ while $j_{\hat{i}}$ is assigned to $a_i$ in the $(p+1)$-th round. Therefore, we have $X_k^{p+1} = X_k^p \cup \{ j_{\hat{k}} \}$ and $X_i^{p+1} = X_i^p \cup \{ j_{\hat{i}} \}$. Since $|J_{p+1}| \leqslant m$, $\{ j_{\hat{k}} \}$ and $\{ j_{\hat{i}} \}$ may be empty, in which case, we assume that $u(j_{\hat{k}}) = u(j_{\hat{i}}) = 0$. Since $a_k$ chooses the job before $a_i$ and all jobs in $J_{p+1}$ are sorted in non-increasing order, $u(j_{\hat{k}}) \geqslant u(j_{\hat{i}})$ always holds no matter whether $\{ j_{\hat{k}} \}$ is empty or not.

Regarding agent $a_i$, as $u_i(X_i^p) \geqslant u_k(X_k^p)$, we have

$$u_i(X_i^{p+1}) \geqslant u_i(X_i^p) \geqslant u_i(X_k^p) = u_i(X_k^{p+1} \setminus \{ j_{\hat{k}} \}).$$

Regarding agent $a_k$, because $u_k(j_{\hat{k}}) \geqslant u_k(j_{\hat{i}})$ and $u_i(X_i^p) \geqslant u_i(X_k^p \setminus \{ j_h \})$ (induction hypothesis),

$$u_k(X_k^{p+1}) = u_k(X_k^p) + u_k(j_{\hat{k}}) \geqslant u_k(X_i^p \setminus \{ j_h \}) + u_k(j_{\hat{i}}) = u_k(X_i^{p+1} \setminus \{ j_h \}).$$

Thus, after the $(p+1)$-th round, $a_i$ and $a_k$ continue not to envy each other for more than one item. By induction, Lemma 9 holds. $\square$

*Proof of Theorem 5.* Since schedule $\mathbf{X}$ returned by Algorithm 5 maximizes social welfare $\sum_{a_i \in A} u_i(X_i)$, $\mathbf{X}$ must be PO. According to Lemma 9, $\mathbf{X}$ is EF1. For time complexity, we have already discussed that computing a maximum $m$-matching can be done in polynomial time. Further, as allocating jobs by types only needs to sort jobs or agents, which can also be done in polynomial time, we finished the proof. $\square$

We note that Algorithm 5 fails to return an EF1 and PO schedule if the agents' utilities are not identical. Actually, the existence of EF1 and PO schedule for this case is left open in Biswas and Barman [2018]; Dror *et al.* [2020]; Wu *et al.* [2021] even when the scheduling constraints degenerate to cardinality constraints.

**Remark 1.** *We noted that the proof of Lemma 9 only uses the ranking of jobs' weight. Therefore, Algorithm 5 is able to return to a feasible schedule that is simultaneously EF1 and PO in the setting where agents value jobs in the same order but the concrete jobs' weight are not known by the algorithm.*

### C.3 1/4-EF1 and PO for general FISP instances

Before giving the proof, we first introduce the formal definition of MaxNSW-schedule which will be used to prove Theorem 4.

**Definition 5** (MaxNSW Schedule). *A feasible schedule $\mathbf{X} = (X_1, \cdots, X_m)$ is called MaxNSW schedule if and only if*

$$\mathbf{X} \in \arg\max_{\mathbf{X}' \in \mathcal{F}} \prod_{i=1}^{m} u_i(X_i')$$

*where $\mathcal{F}$ is the set of all feasible schedules and $\mathbf{X}' = (X_1', \cdots, X_m')$.*

Note that in the standard definition of Nash social welfare maximizing schedule, $\mathbf{X}$ was supposed to be a member of $\arg\max_{\mathbf{X}' \in \mathcal{F}} \left( \prod_{i=1}^{m} u_i(X_i') \right)^{\frac{1}{m}}$. Here, we ignore the power of $\frac{1}{m}$ to simplify the formula.

In this section, we mainly prove Theorem 6 which is the first part of Theorem 4. The proof of Theorem 6 is essentially the same with corresponding one in Wu *et al.* [2021], and we include the proof for completeness.

**Theorem 6.** *Given an arbitrary instance of general FISP, any schedule that maximizes the Nash social welfare is a 1/4-EF1 and PO schedule.*

*Proof.* Given an arbitrary instance of general FISP, let $\mathbf{X} = (X_1, \cdots, X_m)$ be the MaxNSW schedule and let $X_0 = J \setminus \bigcup_{i \in [m]} X_i$. Since any MaxNSW schedule must be a PO schedule, we only prove that $\mathbf{X}$ is a 1/4-EF1 schedule i.e., $\forall i, k \in [m], u_i(X_i) \geqslant \frac{1}{4} u_i(X_k \setminus \{ j_p \}), \exists j_p \in X_k$. Suppose, on the contrary, that there exists $i, k \in [m]$ such that $u_i(X_i) < \frac{1}{4} u_i(X_k \setminus \{ j_p \}), \forall j_p \in X_k$.

Now, we sort all jobs in $X_k$ in non-increasing order according to the value of $u_k(j_p), j_p \in X_k$. Assume that $X_k = \{ j_1, j_2, \cdots \}$ after sorting. Without loss of generality, we assume that $|X_k|$ is an odd number; otherwise, we add a dummy job $j_o$ to $X_k$ such that $u_i(j_o), \forall i \in [m]$. Now we partition $X_k \setminus \{ j_1 \}$ into two subsets $X_k^1, X_k^2$, where $X_k^1 = \{ j_2, j_4, j_6, \cdots \}$ and $X_k^2 = \{ j_3, j_5, j_7, \cdots \}$. Note that $X_k = \{ j_1 \} \cup \{ X_k^1 \} \cup \{ X_k^2 \}$. Note that $u_k(X_k^1) \geqslant u_k(X_k^2)$ and $u_k(X_k^2 \cup \{ j_1 \}) \geqslant u_k(X_k^1)$ since all jobs in $X_k$ are sorted in non-increasing order. Since $u_k(X_k^1) \geqslant u_k(X_k^2)$, we have $u_k(j_1) + u_k(X_k^1) \geqslant u_k(X_k^2)$. Therefore, we have

$$u_k(X_k^d \cup \{ j_1 \}) \geqslant \frac{1}{2} u_k(X_k), \forall d \in \{ 1, 2 \}. \tag{6}$$

Since $u_i(X_i) < \frac{1}{4} u_i(X_k \setminus \{ j_p \}), \forall j_p \in X_k$, we have $u_i(X_i) < \frac{1}{4} u_i(X_k^1 \cup X_k^2)$. Since $X_k$ is a feasible job set, we have $u_i(X_k^1 \cup X_k^2) = u_i(X_k^1) + u_i(X_k^2)$ which implies that either $u_i(X_k^1) \geqslant \frac{1}{2} u_i(X_k^1 \cup X_k^2)$ or $u_i(X_k^1) \geqslant \frac{1}{2} u_i(X_k^1 \cup X_k^2)$. Therefore, we have

$$u_i(X_i) < \frac{1}{4} u_i(X_k^1 \cup X_k^2) \leqslant \frac{1}{2} u_i(X_k^d), \exists d \in \{ 1, 2 \}. \tag{7}$$

Now we construct a new schedule, denoted by $\mathbf{X}' = (X_1', \cdots, X_m')$, where $X_r' = X_r, \forall r \in [m], r \neq i, k$. Let $X_0' = J \setminus \bigcup_{i \in [m]} X_i'$. We discard all jobs in $X_i$, i.e., $X_0' = X_0 \cup X_i$. If $u_i(X_k^1) \geqslant \frac{1}{2} u_i(X_k^1 \cup X_k^2)$, let $X_i' = X_k^1$ and $X_k' = X_k^2 \cup \{ j_1 \}$; otherwise, let $X_i' = X_k^2$ and $X_k' = X_k^1 \cup \{ j_1 \}$. It is easy to see that $\mathbf{X}'$ is a feasible schedule. Note thar all job sets in $\mathbf{X}'$ except $X_0', X_i', X_k'$ are the same as the corresponding job sets in $\mathbf{X}$. Observe that if we can prove that $u_i(X_i') u_k(X_k') > u_i(X_i) u_k(X_k)$, then $\mathbf{X}$ is not a MaxNSW schedule which will contradict our assumption. In the case where $u_i(X_k^1) \geqslant \frac{1}{2} u_i(X_k^1 \cup X_k^2)$, we have $X_i' = X_k^1$. By Equation (6), we have $u_k(X_k') = u_k(X_k^2 \cup \{ j_1 \}) \geqslant \frac{1}{2} u_k(X_k)$. By Equation (7), we have $u_i(X_i') = u_i(X_k^1) > 2 u_i(X_i)$. In the case where $u_i(X_k^1) < \frac{1}{2} u_i(X_k^1 \cup X_k^2)$, we have $X_i' = X_k^2$. By Equation (6), we have $u_k(X_k') = u_k(X_k^1 \cup \{ j_1 \}) \geqslant \frac{1}{2} u_k(X_k)$. By Equation (7), we have $u_i(X_i') = u_i(X_k^2) > 2 u_i(X_i)$. By combining above two cases, we have $u_i(X_i) u_i(X_k) < u_i(X_i') u_k(X_k')$. □

In the following, we show that our proof in Theorem 6 is tight.

**Lemma 10.** *Given an arbitrary instance of general FISP, a MaxNSW schedule can only guarantee 1/4-EF1 and PO.*

*Proof.* To prove Lemma 10, it is sufficient to give an instance such that MaxNSW schedule is exactly 1/4-EF1 schedule and PO. In this instance, all jobs in job set $J$ are rigid and $J$ can be partitioned into two sets $J_L$ and $J_S$. There is only one job in $J_L$ which is very long and has weight 1. There are $\frac{4}{\epsilon}$ jobs in $J_S$ each of which has unit length and weight $\epsilon$. Note that $\frac{4}{\epsilon}$ is assumed to be an even integer number. All jobs in $J_S$ are disjoint and the job in $J_L$ intersects with all jobs in $J_S$. The agent set $A$ contains only two agents, i.e., $|A| = 2$. The instance can be found in Figure 7.

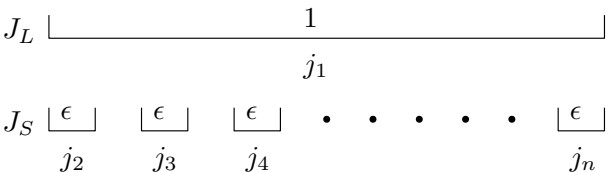

Figure 7: Tight example of MaxNSW schedule for general FISP.

Note that the total weight of jobs in $J_S$ is 4. Let $\mathbf{X} = (X_1, X_2)$ be the schedule, where $X_1 = J_L, X_2 = J_S$. let $\mathbf{X}' = (X_1', X_2')$, where $X_1' = \{ j_2, \cdots, j_{\frac{2}{\epsilon}+1} \}, X_2' = J_S \setminus X_1'$, i.e., $J_S$ is partitioned into two subsets with equal size. Note that $X_0' = J_L$. It is not hard to see that $\mathbf{X}'$ is a MaxNSW schedule. And we have $u_1(X_1) u_2(X_2) = 4, u_1(X_1') u_2(X_2') = 4$. Therefore, $\mathbf{X}$ is a MaxNSW schedule. Note that $u_1(X_2 \setminus \{ j_p \}) = \frac{4-\epsilon}{\epsilon} \cdot \epsilon = 4 - \epsilon, \forall j_p \in X_2$. Therefore, we have

$$\lim_{\epsilon \to 0} \frac{1}{4} u_1(X_2 \setminus \{ j_p \}) = 1 = u_1(X_1), \forall j_p \in X_1.$$

This implies that $\mathbf{X}$ is a 1/4-EF1 schedule. □

## C.4 1/2-EF1 and PO for FISP with ⟨non-identical, unit⟩ instances

In this section, we mainly prove Theorem 7 which is the second part of Theorem 4.

**Theorem 7.** *Given an arbitrary instance of FISP with ⟨non-identical, unit⟩, a MaxNSW schedule is a 1/2-EF1 and PO schedule.*

*Proof.* We show that a feasible schedule $\mathbf{X} = (X_1, \cdots, X_m)$ that maximizes Nash social welfare is simultaneously 1/2-EF1 and PO. Since any MaxNSW schedule must be a PO schedule, we only prove that $\mathbf{X}$ is a 1/2-EF1 schedule Hence, we only show that $\mathbf{X}$ is an 1/2-EF1 schedule, i.e., $\forall i, k \in [m], u_i(X_i) \geqslant \frac{1}{2} \cdot u_i(X_k \setminus \{j\}), \exists j \in X_k$.

We prove by contradiction and assume that there exists $i, k \in [m]$ such that $u_i(X_i) < \frac{1}{2} \cdot u_i(X_k \setminus \{j\}), \forall j \in X_k$. Then, we have

$$u_i(X_i) + u_i(j) < u_i(X_k) - u_i(X_i), \forall j \in X_k. \tag{8}$$

Since $X_i, X_k$ are feasible job set, there is a maximum weighted matching in $G(X_i \cup T, E_i), G(X_k \cup T, E_k)$ with size $|X_i|, |X_k|$, respectively. Let $M_i, M_k$ be the maximum weighted matching in $G(X_i \cup T, E_i), G(X_k \cup T, E_k)$, respectively. An example can be found in Figure 8.

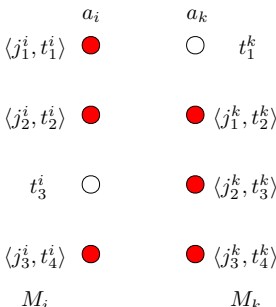

Figure 8: Illustration for $M_i, M_k$. In the above example, we have $X_i = \{j_1^i, j_2^i, j_3^i\}$, $X_k = \{j_1^k, j_2^k, j_3^k\}$ and $T = \{t_1, t_2, t_3, t_4\}$. We have matching $M_i = \{\langle j_1^i, t_1^i \rangle, \langle j_2^i, t_2^i \rangle, \langle j_3^i, t_4^i \rangle\}$ and $M_k = \{\langle j_1^k, t_2^k \rangle, \langle j_2^k, t_3^k \rangle, \langle j_3^k, t_4^k \rangle\}$. Moreover, we have $M_i(J) = X_i, M_i(T) = \{t_1, t_2, t_4\}$ and $M_k(J) = X_k, M_k(T) = \{t_2, t_3, t_4\}$.

For every time slot $t_l \in M_i(T) \cup M_k(T)$, we find the pair $\langle j^i, t_l^i \rangle \in M_i, \langle j^k, t_l^k \rangle \in M_k$. Note that there may exist some time slot $t_l$ such that $t_l$ is only matched in $M_i$ or $M_k$, e.g., time slot $t_1, t_3$ in the example shown in Figure 8. In this case, we add a dummy pair to $M_k$ or $M_i$, e.g., in the example shown in Figure 8, $M_i = M_i \cup \{\langle j_o, t_3^i \rangle\}$, $M_k = M_k \cup \{\langle j_o, t_1^k \rangle\}$ and let $u_i(j_o) = u_k(j_o) = 0, \forall i \in [m]$. For every time slot $t_l \in M_i(T) \cup M_k(T)$, we find the pair $\langle j^i, t_l^i \rangle \in M_i, (j^k, t_l^k) \in M_k$ and define the big pair $[\langle j^i, t_l^i \rangle, \langle j^k, t_l^k \rangle]$ as $(j^i, j^k)$ for convenience. For each pair $(j^i, j^k)$, we define $|(j^i, j^k)|$ as its value, where

$$|(j^i, j^k)| = \frac{u_i(j^k) - u_i(j^i)}{u_k(j^k) - u_k(j^i)}.$$

Note that there may exist two pairs: $\langle j^i, t_l^i \rangle \in M_i, \langle j^k, t_l^k \rangle \in M_k$ such that $u_i(j^k) - u_i(j^i) = 0$ and $u_k(j^k) - u_k(j^i) = 0$. In this case, we have

$$|(j^i, j^k)| = \begin{cases} 0, \text{ if } & u_i(j^k) - u_i(j^i) = 0, u_k(j^k) - u_k(j^i) \neq 0; \\ \infty, \text{ if } & u_i(j^k) - u_i(j^i) \neq 0, u_k(j^k) - u_k(j^i) = 0. \end{cases}$$

Let $\mathcal{P}_+, \mathcal{P}_-$ be the set of all $(j^i, j^k)$ such that $u_i(j^k) - u_i(j^i) > 0$ and $u_i(j^k) - u_i(j^i) \leqslant 0$, respectively. We consider an arbitrary pair $(j_+^i, j_+^k)$ in $\mathcal{P}_+$, i.e., $u_i(j_+^k) - u_i(j_+^i) > 0$. Note that $u_k(j_+^k) - u_k(j_{+0}^i)$ holds; otherwise, we can construct a new feasible schedule by swapping job $j_+^i$ and $j_+^k$ will have larger Nash Social Welfare. This would imply that $\mathbf{X}$ does not maximize the Nash Social Welfare. Let $X_i^+, X_k^+$ be the set of jobs in $X_i, X_k$ that are covered by some pair in

$\mathcal{P}_+$, respectively, i.e., $X_i^+ = \{\, j^i \in X_i \mid \exists (j^i, j^k) \in \mathcal{P}_+ \,\}$ and $X_k^+ = \{\, j^k \in X_k \mid \exists (j^i, j^k) \in \mathcal{P}_+ \,\}$. Notations $X_i^-, X_k^-$ are defined in similar ways. Note that

$$u_i(X_k) - u_i(X_i) = \Big( u_i(X_k^+) - u_i(X_i^+) \Big) + \Big( u_i(X_k^-) - u_i(X_i^-) \Big).$$

Since $u_i(X_k^-) - u_i(X_i^-) \leqslant 0$, we have

$$u_i(X_k) - u_i(X_i) \leqslant u_i(X_k^+) - u_i(X_i^+). \tag{9}$$

Then, we have:

$$\frac{u_i(X_k^+) - u_i(X_i^+)}{u_k(X_k^+)} \geqslant \frac{u_i(X_k^+) - u_i(X_i^+)}{u_k(X_k)} \geqslant \frac{u_i(X_k) - u_i(X_i)}{u_k(X_k)}, \tag{10}$$

where the first inequality is due to $u_i(X_k^+) - u_i(X_i^+) > 0$ and $u_k(X_k) \geqslant u_k(X_k^+)$, the last inequality is due to Equation (9). Now, we define $(g^i, g^k)$ as:

$$(g^i, g^k) = \underset{(j^i, j^k) \in \mathcal{P}_+}{\arg\max} \left\{ |(j^i, j^k)| \right\}.$$

Note that $\mathcal{P}_+ \neq \emptyset$, i.e., there must exist a pair $(j_+^i, j_+^k)$ such that $u_i(j_+^k) - u_i(j_+^i) > 0$ because of $u_i(X_k) > u_i(X_i)$. Since every pair $(j^i, j^k)$ in $\mathcal{P}_+$ has property $u_i(j^k) - u_i(j^i) > 0$ and $u_k(j^k) - u_i(j^i) > 0$, we have:

$$\frac{u_i(g^k) - u_i(g^i)}{u_k(g^k) - u_k(g^i)} \geqslant \frac{u_i(X_k^+) - u_i(X_i^+)}{u_k(X_k^+) - u_k(X_i^+)} \geqslant \frac{u_i(X_k^+) - u_i(X_i^+)}{u_k(X_k^+)}, \tag{11}$$

where the last inequality is due to $u_i(X_k^+) - u_i(X_i^+) > 0$ and $u_k(X_i^+) \geqslant 0$. By combining Equation (10) and Equation (11), we have

$$\frac{u_i(g^k) - u_i(g^i)}{u_k(g^k) - u_k(g^i)} \geqslant \frac{u_i(X_k) - u_i(X_i)}{u_k(X_k)} > \frac{u_i(X_i) + u_i(g^k)}{u_k(X_k)}, \tag{12}$$

where the last inequality is due to Equation (8).

Since $u_i(g^k) - u_i(g^i) > 0$ and $(u_k(g^k) - u_k(g^i) > 0$, we have:

$$\Big( u_i(g^k) - u_i(g^i) \Big) \cdot u_k(X_k) > \Big( u_k(g^k) - u_k(g^i) \Big) \cdot \Big( u_i(X_i) + u_i(g^k) \Big). \tag{13}$$

Holding Equation (13) on our hand, we are ready to prove that $\mathbf{X}$ does not maximize the Nash social welfare. Now, we construct another feasible schedule, denoted by $\mathbf{X}' = (X_1', \cdots, X_m')$. We construct $\mathbf{X}'$ by swapping the job $g^i$ with $g^k$, i.e., $X_o' = X_o, \forall o \in [m]$ and $o \neq i, k$, $X_i' = X_i \cup \{\, g^k \,\} \setminus \{\, g^i \,\}$ and $X_k' = X_k \cup \{\, g^i \,\} \setminus \{\, g^k \,\}$. Note that all job sets in $\mathbf{X}'$ except $X_i', X_k'$ are the same as the corresponding job sets in $\mathbf{X}$. Observe that if we can show that $u_i(X_i')u_k(X_k') > u_i(X_i)u_k(X_k)$, then it implies that $\mathbf{X}$ does not maximize the Nash social welfare. Note that

$$u_i(X_i') = u_i(X_i) + u_i(g^k) - u_i(g^i);$$
$$u_k(X_k') = u_k(X_k) + u_k(g^i) - u_k(g^k).$$

We define $\Gamma$ as follows for convenience:

$$\Gamma = \Big( u_i(g^k) - u_i(g^i) \Big) \cdot u_k(X_k) - \Big( u_k(g^k) - u_k(g^i) \Big) \cdot \Big( u_i(X_i) + u_i(g^k) \Big),$$

where $\Gamma > 0$ because of Equation (13). Then, we have

$$u_i(X_i')u_k(X_k') - u_i(X_i)u_k(X_k) = \Gamma + \Big( u_k(g^k) - u_k(g^i) \Big) \cdot u_i(g^i).$$

Since $u_k(g^k) - u_k(g^i) > 0$ and $\Gamma > 0$, we have $u_i(X_i')u_k(X_k') - u_i(X_i)u_k(X_k) > 0$. Hence $\mathbf{X}$ does not maximize the Nash social welfare which contradicts our assumption. Therefore, $\forall i, k \in [m], u_i(X_i) \geqslant \frac{1}{2} \cdot (X_k \setminus \{\, j \,\}), \exists j \in X_k$. $\qquad \square$

In the following, we show that our proof of Theorem 7 is tight.

**Lemma 11.** *The schedule which maximizes the Nash social welfare can only guarantee 1/2-EF1 and PO for FISP with ⟨non-identical,unit⟩.*

*Proof.* To prove Lemma 11, we give an instance for which a schedule that maximizes the Nash social welfare is an 1/2-EF1 schedule.

We consider the job set $J = \{ j_1, \cdots, j_n, j_{n+1}, \cdots, j_{2n} \}$ which contains $2n$ jobs. All jobs have the same release time $1$ and deadline $n$. Moreover, all jobs have unit processing time. The agent set $A = \{ a_1, a_2 \}$ contains two agents. The utilities matrix is as follows:

|       | $j_1$ | $j_2$ | $\cdots$ | $j_n$ | $j_{n+1}$ | $\cdots$ | $j_{2n}$ |
|-------|-------|-------|----------|-------|-----------|----------|----------|
| $a_1$ | 2     | 2     | $\cdots$ | 2     | 1         | $\cdots$ | 1        |
| $a_2$ | 1     | 1     | $\cdots$ | 1     | 0         | $\cdots$ | 0        |

To find the schedule that maximizes the Nash social welfare, we consider an arbitrary schedule $\mathbf{X} = (X_1, X_2)$ and assume that $X_0 = J \setminus (X_1 \cup X_2)$. We define $J_1 = \{ j_1, \cdots, j_n \}$ and $J_2 = \{ j_{n+1}, \cdots, j_{2n} \}$. Observe that $X_0 = \emptyset$ otherwise $\mathbf{X}$ does not maximize the value of $u_1(X_1) \cdot u_2(X_2)$. We assume that $x$ jobs in $J_1$ are assigned to $a_1$ and $y$ jobs in $J_2$ are assigned to $a_2$, where $0 \leqslant x, y \leqslant n$. Then, we have

$$f(x, y) = u_1(X_1) \cdot u_2(X_2) = (2x + y) \cdot (n - x).$$

To find the maximum value of $f(x, y)$ under the constraints $0 \leqslant x, y \leqslant n$, we compute partial derivative.

$$\begin{cases} \frac{\partial f(x,y)}{\partial x} = 2n - 4x - y = 0 \\ \frac{\partial f(x,y)}{\partial y} = n - x = 0 \end{cases}$$

The solution to the above two equations is $(n, -2n)$. Since the point $(n, -2n) \notin \{ (x, y) \mid 0 \leqslant x, y \leqslant n \}$, the maximum value will be taken at a certain vertex. We can find that the maximum value will be taken at the point $(x, y) = (0, n)$ by computing the value of $f(0, 0), f(0, n), f(n, 0), f(n, n)$.

Hence, we found the schedule $\mathbf{X} = (X_1, X_2)$ maximizes the Nash social welfare, where $X_1 = J_2, X_2 = J_1$. Then, we have $u_1(X_1) = 2n$ and $u_1(X_1 \setminus \{ j \}) = 2(n - 1), \forall j \in X_1$. Then, we have

$$\lim_{n \to +\infty} \frac{u_1(X_1)}{u_1(X_1 \setminus \{ j \})} = \frac{n}{2(n - 1)} = \frac{1}{2}.$$

$\square$

# D  EF1 and IO Scheduling

Lemma 6 shows that PO is very demanding since even if agents have unweighted utilities, EF1 and PO are not compatible. Accordingly, in this section, we will consider the following weaker efficiency criterion – Individual Optimality.

**Definition 6** ($\alpha$-IO schedule). *A feasible schedule $\mathbf{X} = (X_1, \cdots, X_m)$ with $X_0 = J \setminus \bigcup_{i \in [m]} X_i$ is called $\alpha$-approximate individual optimal ($\alpha$-IO) if $u_i(X_i) \geqslant \alpha \cdot u_i(X_0 \cup X_i)$ for all $a_i \in A$, where $\alpha \in (0, 1]$ and when $\alpha = 1$, $\mathbf{X}$ is called IO schedule.*

As we will see, although EF1 and IO are still not compatible for weighted utilities, they are when agents have unweighted utilities.

## D.1  An Impossibility Result

We first show that EF1 and IO are not compatible even for FISP with ⟨identical, rigid⟩, i.e., given an arbitrary instance of FISP with ⟨identical, rigid⟩, there is no algorithm can always find a feasible schedule that is simultaneously EF1 and IO (Lemma 12).

**Lemma 12.** *EF1 and IO are not compatible even for FISP with ⟨identical, rigid⟩.*

*Proof.* To prove Lemma 12, it suffices to consider the instance in Figure 9, and prove the following two claims.

**Claim 3.** *For any IO schedule $\mathbf{X} = (X_1, \cdots, X_m)$, $X_0 = J \setminus \bigcup_{i \in [m]} X_i = \emptyset$.*

We prove this claim by contradiction. If $X_0 \cap J_1 \neq \emptyset$, as $|J_2| = m - 1$, there will be at least one agent, without loss of generality say $a_1$, for whom $X_1 \cap J_2 = \emptyset$. Note that by the design of the instance, $X_1 \cup (X_0 \cap J_1)$ is feasible, and thus by allocating $X_0 \cap J_1$ to $a_1$, $a_1$'s utility strictly increases.

If $X_0 \cap J_2 \neq \emptyset$, as $|J_2| = m - 1$, there will be at least two agents, without loss of generality say $a_1$ and $a_2$, for whom $X_1 \cap J_2 = \emptyset$ and $X_2 \cap J_2 = \emptyset$. Furthermore, as $|J_1| = 4$ one of them gets at most two jobs in $J_1$. Again without loss of generality assume this is agent $a_1$. Accordingly, $u_1(X_1) \leqslant 4$ and by exchanging $X_1$ with one job in $X_0 \cap J_2$, $a_1$'s utility strictly increases.

**Claim 4.** *For any EF1 schedule $\mathbf{X} = (X_1, \cdots, X_m)$, $X_0 = J \setminus \bigcup_{i \in [m]} X_i \neq \emptyset$.*

We note that the only possible and feasible schedule $\mathbf{X}$ such that $X_0 = \emptyset$ is that some agent, say $a_1$, gets entire $J_1$ and every other agent gets one job in $J_2$. Then to prove this claim, it suffices to prove $\mathbf{X}$ cannot be EF1. It is not hard to check that under $\mathbf{X}$, for any agent $a_i$ with $i \geqslant 2$ and any job $j \in X_1$,

$$u_i(X_i) = 6 - \epsilon < u_i(X_1 \setminus \{j\}) = 6.$$

That is all $a_i$ envies $a_1$ for more than one item.

Combing the above two claims, we complete the proof of Lemma 12. □

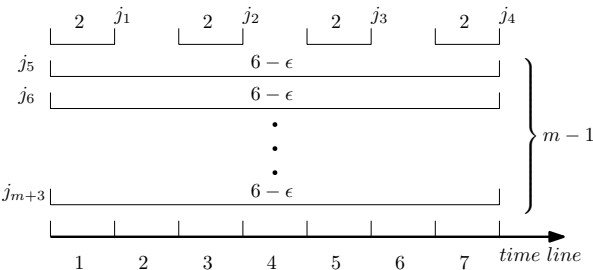

Figure 9: Instance for Lemma 12. There are $|A| = m$ agents and $|J| = m + 3$ jobs with $m \geqslant 2$. Job set $J$ can be partitioned as $J_1 \cup J_2$ with $J_1 = \{j_1, j_2, j_3, j_4\}$ and $J_2 = \{j_5, \cdots, j_{m+3}\}$. Each job in $J_1$ has unit processing time with weight 2 and each job in $J_2$ has processing time 7 with weight $6 - \epsilon$. All jobs are rigid such that $j_i \in J_1$ needs to occupy the entire time slot $2i - 1$, and $j \in J_2$ occupies the entire time period from 1 to 7.

## D.2 A Polynomial-time Algorithm for FISP with ⟨unweighted, rigid⟩

In the following, we design a polynomial-time algorithm to compute a schedule that is EF1 and IO for any instance of FISP with ⟨unweighted, rigid⟩.

**Theorem 8.** *Given an arbitrary instance of FISP with ⟨unweighted, rigid⟩, Algorithm 6 returns a feasible schedule that is simultaneously EF1 and IO in polynomial time.*

Let $\mathbf{X} = (X_1, \cdots, X_m)$ be the schedule returned by Algorithm 6 and let $X_0 = J \setminus \bigcup_{i \in [m]} X_i$. Suppose that all agents receive a job at every round of first $L$ rounds of Algorithm 6, i.e., in the $(L+1)$-th round, $\exists i \in [m]$ such that $a_i$ receives nothing. Note that $L \leqslant n$, where $n$ is the number of jobs. Let $j_i^l, 1 \leqslant i \leqslant m, 1 \leqslant l \leqslant L$, be the job assigned to agent $a_i$ in the $l$-th round of Algorithm 6. Let $r_i^l, d_i^l$ be the release time and deadline of job $j_i^l$. Let $X_i^l$ be the job set that is assigned to agent $a_i$ after the $l$-th round of Algorithm 6.

**Lemma 13.** $d_1^l \leqslant d_2^l \leqslant \cdots \leqslant d_m^l, \forall l \in [L]$.

---

**Algorithm 6.** Earliest Deadline First + Round-Robin

---

**Input:** Agent set $A$ and job set $J$.
**Output:** EF1 schedule $\mathbf{X} = (X_1, \cdots, X_m)$
1: Sort all jobs by their deadline in non-decreasing order.
2: $X_1 = X_2 = \cdots = X_m = \emptyset$.
3: $i = 1, k = 1$. // The index.
4: **for all** $j_k \in J$ **do**
5:     **if** $X_i \cup \{ j_k \}$ is a feasible job set **then**
6:         $X_i = X_i \cup \{ j_k \}$.
7:         $J = J \setminus \{ j_k \}$.
8:         $i = (i + 1) \mod m$.
9:     **else**
10:         $i = i \mod m$.
11:     **end if**
12: **end for**
13: $X_0 = J \setminus \bigcup_{i \in [m]} X_i$.

---

*Proof.* We consider two agents $a_i, a_k$ such that $1 \leqslant i < k \leqslant m$. Note that $j_i^l, j_k^l$ must exist since, in the first $L$ rounds, all agents receive a job. We prove by induction.

**Base case and induction hypothesis.** In the base case where $l = 1$, it is straightforward to see that $d_i^1 \leqslant d_k^1$, otherwise $j_k^1$ will be assigned to agent $a_i$ in the first round of Algorithm 6. Now, we have induction hypothesis $d_i^l \leqslant d_k^l$.

We need to prove $d_i^{l+1} \leqslant d_k^{l+1}$. We prove by contradiction and assume that $d_i^{l+1} > d_k^{l+1}$. Since agent $a_i$ chooses $j_i^{l+1}$ instead of $j_k^{l+1}$ in the $(l + 1)$-th round, we know that $X_i^l \cup \{ j_k^{l+1} \}$ is a not feasible job set which implies that $r_k^{l+1} \leqslant d_i^l$. By induction hypothesis, we have $r_k^{l+1} \leqslant d_i^l \leqslant d_k^l$. This implies that $X_k^l \cup \{ j_k^{l+1} \}$ is not a feasible job set. This contradicts our assumption. Thus, $d_i^{l+1} \leqslant d_k^{l+1}$. $\qquad\square$

**Lemma 14.** $d_m^l \leqslant d_1^{l+1}, \forall l \in [L - 1]$. *Moreover,* $d_m^L \leqslant d_1^{L+1}$ *if* $j_1^{L+1}$ *exists.*

*Proof.* Let $a_i, a_k$ be two agents such that $1 \leqslant i < k \leqslant m$. We assume that $j_1^{L+1}$ exists and prove that the lemma holds for all $l \in [L]$. We prove by contradiction.

In the base case where $l = 1$, it is not hard to see that $d_m^l \leqslant d_1^2$; otherwise $a_m$ will choose $j_1^2$ in the first round. Now, we have induction hypothesis $d_m^{l-1} \leqslant d_1^l$.

Suppose, towards to a contradiction, that there exist $l \in [L]$ such that $d_m^l > d_1^{l+1}$. In the $l$-th round, agent $a_m$ selects $j_m^l$ instead of $j_1^{l+1}$ because $X_m^{l-1} \cup \{ j_1^{l+1} \}$ is not a feasible job set; otherwise $a_m$ will select $j_1^{l+1}$. Since $X_m^{l-1} \cup \{ j_1^{l+1} \}$ is not a feasible job set, we have $r_1^{l+1} \leqslant d_m^{l-1}$. By induction hypothesis, we have $d_m^{l-1} \leqslant d_1^l$. Therefore, we have $r_1^{l+1} \leqslant d_1^l$ which implies that $X_1^l \cup \{ j_1^{l+1} \}$ is not a feasible job set. This contradicts our assumption. $\qquad\square$

**Lemma 15.** $|X_i| - |X_k| \in \{ -1, 0, 1 \}, \forall i, k \in [m]$.

*Proof.* We consider the $(L + 1)$-th round of Algorithm 6 in which $\exists f \in [m]$ such that $a_f$ receives nothing in this round. Let $J_f^L$ be the set of remaining jobs in $J$ after $a_f$ chooses in the $(L + 1)$-th round. We consider the agent $a_k$ such that $1 \leqslant f \leqslant k < m$. Since $a_f$ receives nothing, we have $r_j \leqslant d_f^L, \forall j \in J_f^L$. According to Lemma 13, we have $d_f^L \leqslant d_k^L$. Then, we have $r_j \leqslant d_f^L \leqslant d_k^L, \forall j \in J_f^L$ which implies that agent $a_k$ also receives nothing in this round. Therefore, $a_m$ must receive nothing in the $(L + 1)$-th round because there exist an agent that does not receive job in the $(L + 1)$-th round. Let $J_m^L$ be the remaining jobs in $J$ before $a_m$ chooses in the $(L + 1)$-th round. Since $a_m$ receives nothing in the $(L + 1)$-th round, we have $r_j \leqslant d_m^L, \forall j \in J_m^L$. According to Lemma 14, we have $d_m^L \leqslant d_1^{L+1}$ if $j_1^{L+1}$ exists. Therefore, we have $r_j \leqslant d_1^{L+1}, \forall j \in J_m^L$. Thus, $a_1$ will receive nothing in the $(L + 2)$-th round. Now, we consider an arbitrary agent $a_h, 1 \leqslant h \leqslant m$, it is straightforward

to see that if $a_h$ receives nothing in $(L+1)$-th round, then $a_h$ will receive nothing in any $L'$-th round, where $L+1 < L'$. Note that, in the $(L+1)$-th round, there may exist many agents that receive nothing. Without loss of generality, we assume that $a_f$ is the agent with the smallest index who receives nothing in the $(L+1)$-th round. Therefore, we have

$$|X_i| = \begin{cases} L, & \forall f \leqslant i \leqslant m; \\ L+1, & \forall 1 \leqslant i < f. \end{cases}$$

Thus, we have $|X_i| - |X_k| \in \{-1, 0, 1\}, \forall i, k \in [m]$. $\qquad\square$

**Lemma 16.** $u_i(X_i) \geqslant u_i(X_0 \cup X_i), \forall i \in [m]$.

We will use the optimal argument for classical interval scheduling to prove Lemma 16. We restate the problem and optimal argument for completeness.

In classical interval scheduling, we are given a set of intervals $\mathcal{I} = \{I_1, I_2, \cdots, I_n\}$. Each interval is associated with a release time and a deadline. A set of intervals $\mathcal{I}'$ is called a compatible set if and only if, for every two intervals $I_k, I_h \in \mathcal{I}'$, $I_k, I_h$ do not intersect. The goad is to find the compatible set with the maximum size. This problem can be easily solved by *Earlier Deadline First* (EDF) Kleinberg and Tardos [2006].

*Proof.* We consider an arbitrary agent $a_i$. We prove by constructing an instance of classical interval scheduling problem. Let $\mathcal{I} = X_0 \cup X_i$. Let ALGE be the interval set selected by EDF algorithm. Observe that if we can prove that ALGE $= X_i$, then it implies that $u_i(X_i) \geqslant u_i(X_0 \cup X_i)$ since ALGE is the optimal solution. Suppose that ALGE $= \{j'_1, j'_2, \cdots, j'_h\}$ and assume that the interval is added to ALGE by EDF algorithm in this order. Suppose that $X_i = \{j_1, j_2, \cdots, j_k\}$ and assume that the job is added to $X_i$ by Algorithm 6 in this order. Note that $|X_i| \leqslant |$ALGE$|$ since ALGE is the compatible set with the maximum size.

We prove by comparison. Assume that ALGE and $X_i$ become different from the $R$-th element, i.e., $j_l = j'_l, \forall l \in [R-1]$ and $j_R \neq j'_R$. This implies that $j'_R$ instead of $j_R$ is the job with the smallest deadline in $X_0 \cup X_i \setminus \{j_1, \cdots, j_{R-1}\}$ to make $\{j_1, \cdots, j_{R-1}\} \cup \{j'_R\}$ be compatible. Note that both $\{j_1, \cdots, j_{R-1}\} \cup \{j'_R\}$ and $\{j_1, \cdots, j_{R-1}\} \cup \{j_R\}$ are feasible. Since $j'_R$ is left to charity, there is no agent takes it away. Therefore, Algorithm 6 will assign $j'_R$ instead of $j_R$ to $a_i$. Hence, we proved $X_i \subseteq$ ALGE. It is easy to see that there is no interval $j'_u \in$ ALGE such that $j'_u \notin X_i$ which implies that $X_i =$ ALGE. $\qquad\square$

Now, we are ready to prove Theorem 8.

*Proof of Theorem 8.* According to Lemma 15, we know that the feasible schedule $\mathbf{X}$ returned by Algorithm 6 is an EF1 schedule. According to Lemma 16, $\mathbf{X}$ is also an IO schedule. Hence, Algorithm 6 returns a feasible schedule that is simultaneously EF1 and IO.

Now, we prove the running time. Line 1 requires running time $O(n \log n)$, where $n$ is the number of jobs. Line 4-13 requires running time $O(n)$. Hence, the running time of Algorithm 6 can be bounded by $O(n \log n)$. $\qquad\square$

Now, we show an instance that Algorithm 6 returns a schedule that is not PO schedule. See Figure 10. By applying Algorithm 6 to the instance in Figure 10, let $\mathbf{X}$ be the returned schedule. Then, we have $\mathbf{X} = (X_1, X_2)$, where $X_1 = \{j_1, j_4\}$, $X_2 = \{j_2, j_5\}$ and $X_0 = \{j_3, j_6\}$. But a possible PO schedule is $\mathbf{X}' = (X'_1, X'_2)$, where $X'_1 = \{j_1, j_4, j_5\}$, $X'_2 = \{j_2, j_6\}$ and $X'_0 = \{j_3\}$.

### D.3 A polynomial time algorithm for FISP with ⟨unweighted, flexible⟩

Note that Algorithm 6 can be modified to run on instances of FISP with ⟨unweighted, flexible⟩. But this modified algorithm fails to return an IO schedule. This is not surprising as it has been proved in Garey and Johnson [1979] that even with a single machine, finding an IO schedule is NP-hard. Fortunately, the modified algorithm still runs in polynomial time and always returns a schedule that is EF1 and 1/2-IO.

Before giving the round-robin algorithm, we first re-state the following classical scheduling problem.

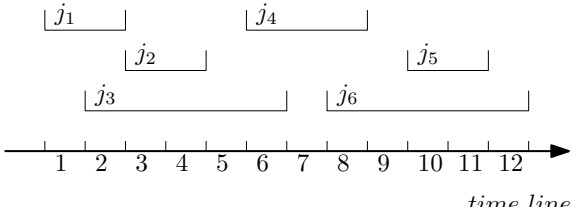

Figure 10: Instance for which Algorithm 6 fails to return a PO schedule. In the above instance, we have $J = \{\, j_1, j_2, j_3, j_4, j_5, j_6 \,\}, A = \{\, a_1, a_2 \,\}$. The job windows are $T_1 = \{\, 1, 2 \,\}, T_2 = \{\, 3, 4 \,\}, T_3 = \{\, 2, 3, 4, 5, 6 \,\}, T_4 = \{\, 6, 7, 8 \,\}, T_5 = \{\, 10, 11 \,\}, T_6 = \{\, 8, 9, 10, 11, 12 \,\}$, respectively.

**Scheduling to find the maximum compatible job set** We are given a job set $J$ which contains $n$ jobs, i.e., $J = \{\, j_1, j_2, \cdots, j_n \,\}$, with each job regraded as a tuple, i.e., $j_i = (r_i, p_i, d_i), i \in [n], 1 \leqslant p_i \leqslant d_i - r_i + 1$, where $r_i, p_i, d_i$ are the release time, processing time and deadline, respectively. There is one machine which is used to process jobs. A subset $J'$ of jobs is called *compatible job set* if and only if all jobs in $J'$ can be finished without preemption before their deadlines. The objective is to find a compatible job set with the maximum size.

The above scheduling problem is the optimization version of the scheduling problem SEQUENCING WITH RELEASE TIMES AND DEADLINES, which is strongly NP-complete Garey and Johnson [1979]. In Bar-Noy *et al.* [2001], they give an $\frac{(m+1)^m}{(m+1)^m - m^m}$-approximation algorithm for $m$ identical machines case. In particular, the approximation ratio is $2$ when $m = 1$. We restate the greedy algorithm for completeness (Algorithm 7).

---

**Algorithm 7.** 2-approximation for scheduling problem on single machine.

1: $\mathsf{ALGC} = \emptyset$.
2: $J^* = J$.
3: $D = 0$.
4: **while** $J^* \neq \emptyset$ **do**
5:     $J^* = \emptyset$. // reset $J^*$.
6:     **for** every job $j \in J$ **do**
7:         **if** $d_j \leqslant \max\{D, r_j\} + p_j$ **then**
8:             $J^* = J^* \cup \{\, j \,\}$.
9:         **end if**
10:     **end for**
11:     $j^* = \arg\min_{j \in J^*}\{\max\{D, r_j\} + p_j\}$.
12:     Schedule job $j^*$ at time slot $\max\{D, r_j\}$.
13:     $D = \max\{D, r_{j^*}\} + p_{j^*}$.
14:     $\mathsf{ALGC} = \mathsf{ALGC} \cup \{\, j^* \,\}$.
15: **end while**

---

**Theorem 9.** *A schedule that is simultaneously EF1 and 1/2-IO exists and can be found in polynomial time for all instance of FISP $\langle$unweighted, flexible$\rangle$.*

Now, we are ready to give the algorithm (Algorithm 8) for instance of FISP $\langle$unweighted, flexible$\rangle$.

Let $\mathbf{X} = (X_1, \cdots, X_m)$ be the schedule returned by Algorithm 8 and $X_0 = J \setminus \bigcup_{i \in [m]} X_i$. We first show that $\mathbf{X}$ is an 1/2-IO schedule and then prove that $\mathbf{X}$ is an EF1 schedule.

**Lemma 17.** $u_i(X_i) \geqslant \frac{1}{2} \cdot u_i(X_i \cup X_0), \forall a_i \in A$.

*Proof.* We consider an arbitrary agent $a_i \in A$ and the job set $X_i \cup X_0$. Let $\mathsf{ALGC}$ be the set of jobs selected from $X_0 \cup X_i$ by Algorithm 7. Let $\mathsf{OPTC}$ be the set of jobs selected by the optimal algorithm. Since Algorithm 7 is a 2-approximation algorithm, we have $|\mathsf{ALGC}| \geqslant \frac{1}{2} \cdot |\mathsf{OPTC}|$. Observe that if we can prove that $\mathsf{ALGC} = X_i$, then we have $u_i(X_i) \geqslant \frac{1}{2} \cdot u_i(X_i \cup X_0)$ since $\mathsf{ALGC}$ has the size at least half of the optimal solution. Let $\mathsf{ALGC} = \{\, j_1, j_2, \cdots, j_k \,\}$ and assume that the jobs are added to the solution by Algorithm 7 in this order. Let $X_i = \{\, j'_1, j'_2, \cdots, j'_r \,\}$ and assume that the jobs

**Algorithm 8.** Round-Robin for FISP ⟨unweighted, flexible⟩

---

**Input:** Agent set $A$ and job set $J$.
**Output:** EF1 schedule $\mathbf{X} = (X_1, \cdots, X_m)$.
1: $X_1 = \cdots = X_m = \emptyset$.
2: $J_1^* = \cdots = J_m^* = J$.
3: $D_1 = \cdots = D_m = 0$.
4: $i = 1$. // The index.
5: **while** there is a $J_i^* \neq \emptyset$ **do**
6:     **for all** $a_i \in A$ **do**
7:         $J_i^* = \emptyset$. // reset $J_i^*$.
8:         **for** every job $j \in J$ **do**
9:             **if** $d_j \leqslant \max\{D_i, r_j\} + p_j$ **then**
10:                $J_i^* = J_i^* \cup \{\, j \,\}$.
11:             **end if**
12:         **end for**
13:         $j_i^* = \underset{j \in J_i^*}{\arg\min}\{\max\{D_i, r_j\} + p_j\}$.
14:         $X_i = X_i \cup \{\, j_i^* \,\}$.
15:         Schedule job $j_i^*$ at $\max\{D_i, r_j\}$.
16:         $D_i = \max\{D_i, r_{j_i^*}\} + p_{j_i^*}$.
17:         $J = J \setminus \{\, j_i^* \,\}$.
18:     **end for**
19: **end while**
20: $X_0 = J \setminus \bigcup_{i \in [m]} X_i$.

---

are added to $X_i$ by Algorithm 8 in this order. We prove by comparison. Assume that ALGC and $X_i$ become different from the $R$-th element, i.e., $j_l = j_l', \forall l \in [R-1]$ and $j_R \neq j_R'$. Assume that the completion time of $j_{R-1}$ is $D_{R-1}$. Then, we have

$$j_R' \neq j_R = \underset{j \in J_R^*}{\arg\min}\{\max\{D_{R-1}, r_j\} + p_j\},$$

where $J_R^* \subseteq \mathcal{J}_r = X_0 \cup X_i \setminus \{\, j_1, \cdots, j_R \,\}$ is a set of jobs which can be feasibly scheduled after $j_R$, i.e.,

$$J_R^* = \{\, j \in \mathcal{J}_r \mid \max\{D_{R-1}, r_j\} + p_j \leqslant d_j \,\}.$$

Note that $j_R' \in J_R^*$. Since $j_R'$ instead of $j_R$ is assigned to agent $a_i$ in a certain round, we know that $j_R$ must be assigned to a certain agent before agent $a_i$ chooses, i.e., $j_R \in X_k, \exists k \in [m]$. This contradicts our assumption since $j_R \in X_0$.

$\square$

Let $J_i^l$ be the job set $J_i^*$ for agent $a_i \in A$ in the $l$-th round, where $1 \leqslant i \leqslant m$ and $1 \leqslant l \leqslant L$. Suppose that in first $L$-th rounds of Algorithm 8, $J_i^* \neq \emptyset, \forall i \in [m]$, i.e., in the $(L+1)$-th round, $\exists i \in [m]$ such that $J_i^* = \emptyset$. Let $D_i^l$ be the parameter $D_i$ in Algorithm 8 for agent $a_i \in A$ at the end of the $l$-th round, where $1 \leqslant i \leqslant m$ and $1 \leqslant l \leqslant L$.

**Lemma 18.** $D_1^l \leqslant D_2^l \leqslant \cdots \leqslant D_m^l, \forall l \in [L]$. *Moreover, we have* $D_m^l \leqslant D_1^{l+1}, \forall l \in [L-1]$, *and* $D_m^L \leqslant D_1^{L+1}$ *if* $J_1^{L+1} \neq \emptyset$.

*Proof.* We first prove $D_1^l \leqslant D_2^l \leqslant \cdots \leqslant D_m^l, \forall l \in [L]$. Let $a_k, a_h \in A$ be two agents, where $1 \leqslant k < h \leqslant n$. We only need to prove $D_k^l \leqslant D_h^l, \forall l \in [L]$. We prove by induction. In the base case where $l = 1$, $D_k^1 \leqslant D_h^1$ obviously holds since agent $a_k$ chooses the job before $a_h$. Now, we have induction hypothesis $D_k^l \leqslant D_h^l$ and we need to prove $D_k^{l+1} \leqslant D_h^{l+1}$. We prove by contradiction and assume that $D_k^{l+1} > D_h^{l+1}$. Let $j_k^{l+1}, j_h^{l+1}$ be the jobs that are selected by agent $a_k, a_h$ in the $(l+1)$-round of Algorithm 8, respectively. Hence, we have

$$D_k^{l+1} = \max\{D_k^l, r_{j_k^{l+1}}\} + p_{j_k^{l+1}},$$
$$D_h^{l+1} = \max\{D_h^l, r_{j_h^{l+1}}\} + p_{j_h^{l+1}}.$$

By induction hypothesis $D_k^l \leqslant D_h^l$, we have

$$\max\{D_k^l, r_{j_h^{l+1}}\} + p_{j_h^{l+1}} \leqslant \max\{D_h^l, r_{j_h^{l+1}}\} + p_{j_h^{l+1}} \leqslant d_{j_h^{l+1}}.$$

This implies that $j_h^{l+1} \in J_k^{l+1}$. Since

$$\max\{D_k^l, r_{j_h^{l+1}}\} + p_{j_h^{l+1}} \leqslant D_h^{l+1} < D_k^{l+1},$$

we have

$$\max\{D_k^l, r_{j_h^{l+1}}\} + p_{j_h^{l+1}} < \max\{D_k^l, r_{j_k^{l+1}}\} + p_{j_k^{l+1}}.$$

This implies that $j_h^{l+1}$ instead of $j_k^{l+1}$ will be chosen by agent $a_k$ in the $(l+1)$-th round of Algorithm 8. This contradicts our assumption.

We assume that $J_1^{L+1} \neq \emptyset$ and prove that $D_m^l \leqslant D_1^{l+1}, \forall l \in [L]$. We prove $D_m^l \leqslant D_1^{l+1}$ holds for any $1 \leqslant l \leqslant L$. Note that $D_r^0 = 0, \forall r \in [m]$. We prove by induction. In the base case where $l = 1$, if $D_m^1 > D_1^2$, $a_m$ will choose $j_1^2$ instead of $j_m^1$ in the first round. Now, we have induction hypothesis $D_m^l \leqslant D_1^{l+1}$ and we need to prove that $D_m^{l+1} \leqslant D_1^{l+2}$ holds. We prove by contradiction and assume that $D_m^{l+1} > D_1^{l+2}$. Let $j_m^{l+1}, j_1^{l+2}$ be the jobs that are selected by agent $a_m, a_1$ in the $(l+1), (l+2)$-th round of Algorithm 8, respectively. Note that in the case where $l = L - 1$, there always exists a job $j_m^{l+2}$ since $\{j_m^{L+1}\} \neq \emptyset$. Hence, we have

$$D_m^{l+1} = \max\{D_m^l, r_{j_m^{l+1}}\} + p_{j_m^{l+1}},$$
$$D_1^{l+2} = \max\{D_1^{l+1}, r_{j_1^{l+2}}\} + p_{j_1^{l+2}}.$$

By induction hypothesis $D_m^l \leqslant D_1^{l+1}$, we have

$$\max\{D_m^l, r_{j_1^{l+2}}\} + p_{j_1^{l+2}} \leqslant \max\{D_1^{l+1}, r_{j_1^{l+2}}\} + p_{j_1^{l+2}} \leqslant d_{j_1^{l+2}}.$$

This implies that $j_1^{l+2} \in J_m^{l+1}$. Since

$$\max\{D_m^l, r_{j_1^{l+2}}\} + p_{j_1^{l+2}} \leqslant D_1^{l+2} < D_m^{l+1},$$

we have

$$\max\{D_m^l, r_{j_1^{l+2}}\} + p_{j_1^{l+2}} < \max\{D_m^l, r_{j_m^{l+1}}\} + p_{j_m^{l+1}}.$$

This implies that $j_1^{l+2}$ instead of $j_m^{l+1}$ will be chosen by agent $a_m$ in the $(l+1)$-th round of Algorithm 8. This contradicts our assumption. $\qquad\square$

**Lemma 19.** $J_1^l \supseteq J_2^l \supseteq \cdots \supseteq J_m^l, \forall l \in [L+1]$. *Moreover, we have $J_m^l \supseteq J_1^{l+1}, l \in [L]$.*

*Proof.* We first prove that $J_1^l \supseteq J_2^l \supseteq \cdots \supseteq J_m^l, \forall l \in [L]$. Let $a_k, a_h \in A$ be two agents, where $1 \leqslant k < h \leqslant n$. We only need to prove $J_k^l \supseteq J_h^l$. To prove $J_h^l \subseteq J_k^l$, we consider an arbitrary job $j \in J_h^l$ and show that $j \in J_k^l$. Note that $J_k^l, J_h^l \neq \emptyset, \forall l \in [L]$. Let $\mathcal{J}_s^k, \mathcal{J}_s^h$ be the set of jobs that are already assigned to the agents before agent $a_k$ and $a_h$ select, respectively. Note that $\mathcal{J}_s^k \subseteq \mathcal{J}_s^h$. According to Algorithm 8, we have

$$J_k^l = \{ j \in (J \setminus \mathcal{J}_s^k) \mid d_j \leqslant \max\{D_k^{l-1}, r_j\} + p_j \},$$
$$J_h^l = \{ j \in (J \setminus \mathcal{J}_s^h) \mid d_j \leqslant \max\{D_h^{l-1}, r_j\} + p_j \}.$$

Since $\mathcal{J}_s^k \subseteq \mathcal{J}_s^h$, we have $J \setminus \mathcal{J}_s^k \supseteq J \setminus \mathcal{J}_s^h$. Now we consider an arbitrary job $j \in J_h^l$ and show that $j$ is also a member of $J_k^l$. Since $j \in (J \setminus \mathcal{J}_s^h)$ and $J \setminus \mathcal{J}_s^k \supseteq J \setminus \mathcal{J}_s^h$, we have $j \in (J \setminus \mathcal{J}_s^k)$. Since $j \in J_h^l$, we have

$$d_j \leqslant \max\{D_h^{l-1}, r_j\} + p_j.$$

According to Lemma 18, $D_h^{l-1} \geqslant D_k^{l-1}$, we have

$$d_j \leqslant \max\{D_k^{l-1}, r_j\} + p_j,$$

which implies that $j \in J_k^l$. Now we consider the case where $l = L + 1$. Note that in the $(L+1)$-th round of Algorithm 8, $\exists i \in [m]$ such that $J_i^{L+1} = \emptyset$. Now we prove that $J_k^{L+1} \supseteq J_h^{L+1}$. If

$J_h^{L+1} = \emptyset$, then we are done. Hence, we assume that $J_h^{L+1} \neq \emptyset$. By a similar argument, a job $j \in J_h^{L+1}$ has the property $d_j \leqslant \max\{D_h^L, r_j\} + p_j$. Then we have $d_j \leqslant \max\{D_k^L, r_j\} + p_j$ holds since $D_k^L \leqslant D_h^L$. Then we have $j \in J_k^{L+1}$.

Now, we prove that $J_m^l \supseteq J_1^{L+1}, \forall l \in [L]$. Note that it is possible that $J_1^{L+1} = \emptyset$. In this case $J_m^L \supseteq J_1^{L+1}$ trivially holds. Hence, we assume that $J_1^{L+1} \neq \emptyset$. To prove $J_m^l \supseteq J_1^{l+1}$, we consider an arbitrary job $j \in J_1^{l+1}$ and show that $j \in J_m^l$. Let $\mathcal{J}_s^m, \mathcal{J}_s^1$ be the set of jobs that are already assigned to the agents before agent $a_m$ and $a_1$ select in the $l, (l+1)$-th round of Algorithm 8, respectively. Note that $\mathcal{J}_s^m \subseteq \mathcal{J}_s^1$. According to Algorithm 8, we have

$$J_m^l = \{\, j \in (J \setminus \mathcal{J}_s^m) \mid d_j \leqslant \max\{D_m^{l-1}, r_j\} + p_j \,\},$$
$$J_1^{l+1} = \{\, j \in (J \setminus \mathcal{J}_s^1) \mid d_j \leqslant \max\{D_1^l, r_j\} + p_j \,\}.$$

Since $\mathcal{J}_s^m \subseteq \mathcal{J}_s^1$, we have $J \setminus \mathcal{J}_s^m \supseteq J \setminus \mathcal{J}_s^1$. Now we consider an arbitrary job $j \in J_1^{l+1}$ and show that $j \in J_m^l$. Since $j \in J_1^{l+1}$, we have

$$d_j \leqslant \max\{D_1^l, r_j\} + p_j.$$

According to Lemma 18, we have $D_m^{l-1} \leqslant D_1^l$. Then, we have

$$\max\{D_1^l, r_j\} + p_j \geqslant \max\{D_m^{l-1}, r_j\} + p_j.$$

Hence, we have $d_j \leqslant \max\{D_m^{l-1}, r_j\} + p_j$ which implies that $j \in J_m^l$. □

**Lemma 20.** $|X_i| - |X_k| \in \{-1, 0, 1\}, \forall i, k \in [m]$.

*Proof.* We consider the $(L+1)$-th round of Algorithm 8 in which there exists an agent $a_i \in A$ such that $a_i$ does not choose any jobs for the first time. Note that there may exist many agents that do not choose any job for the first time in $(L+1)$-th round. We assume that $a_f$ is the first agent that chooses nothing in the $(L+1)$-th round. Since $a_f$ chooses nothing, we have $J_f^{L+1} = \emptyset$. According to Lemma 19, we have $J_i^{L+1} = \emptyset, \forall f \leqslant i \leqslant n$. Moreover, we have $J_i^{L'} = \emptyset, \forall i \in [m]$ and $L + 1 < L'$. Therefore, we have

$$|X_i| = \begin{cases} L, & \forall f \leqslant i \leqslant m; \\ L+1, & \forall 1 \leqslant i < f. \end{cases}$$

This implies that Lemma 20 holds. □

Now we are ready to prove Theorem 9.

*Proof of Theorem 9.* According to Lemma 20 and Lemma 17, we know that Algorithm 8 will return a feasible schedule that is simultaneously EF1 and 1/2-IO.

Now we bound the running time. According to Lemma 20 and Lemma 17, we know that line 5-20 will be run at most $\lceil \frac{n}{m} \rceil$ times, where $n$ is the number of jobs and $m$ is the number of agents. In each while loop, line 6-18 will be run at most $m$ times. In each for loop, line 8-12 will be run at most $n$ times and the running time of line 13 can be bounded by $O(n)$. Hence, we have the running time of Algorithm 8 $O(\lceil \frac{n}{m} \rceil \cdot m \cdot (n^2 + n)) = O(mn^3)$. □

# E  Missing Discussions in Experiments in Section 5

We now empirically test the performance of Algorithm 4 when jobs are rigid, comparing it against a simple Round-Robin algorithm. In this simple Round-Robin algorithm, all jobs are sorted by their deadlines in non-decreasing order. Then every agent picks a job in round-robin manner. Finally, every agent computes the compatible intervals with the maximum weight and all the remaining jobs will be assigned to charity. The formal description can be found in Algorithm 9 with $J' = J$ and $A' = A$. For the experiments, we have implemented both Algorithm 4 and the above round-robin algorithm.

We run our experiments on three job sets with different sizes: 100 (Figure 1 (a)), 500 (Figure 1 (b)) and 1000 (Figure 1 (c)). The release time and deadline of each job is uniformly randomly sampled from the interval [0,50]. For each job set, we further set up three subgroups according to the agents'

---

**Algorithm 9.** Round-Robin (RR)

---

**Input:** Agent set $A'$ and job set $J'$.
**Output:** EF1 schedule $\mathbf{X} = (X_1, \cdots, X_{|A'|})$
  1: Sort all jobs by their deadline in non-decreasing order.
  2: $X_1 = X_2 = \cdots = X_{|A'|} = \emptyset$.
  3: $i = 1, k = 1$. // The index.
  4: **for all** $j_k \in J'$ **do**
  5:    **for all** $a_i \in A'$ **do**
  6:       **if** $k \mod |A'| = i$ **then**
  7:          $X_i = X_i \cup \{ j_k \}$.
  8:       **end if**
  9:    **end for**
 10: **end for**
 11: $i = 1$. // Reset the index
 12: **for all** $X_i$ **do**
 13:    Let $X_i' \subseteq X_i$ be the compatible job set with the maximum weight for agent $a_i$.
 14:    $X_i = X_i'$.
 15: **end for**
 16: $X_0 = J \setminus \bigcup_{i \in [|A'|]} X_i$.

---

utility of every job: (i) the utility gain is sampled uniformly randomly from [1,20]; (ii) the utility gain follows Poisson Distribution with means 50; (iii) the utility gain follows Normal Distribution with means 25 and variance 10. For each subgroup, we further set up three subsubgroups according to the size of agent set: 5, 10 and 15.

In total, our experiment contains $3 \times 3 \times 3$ groups. For each group, we run Algorithm 4 and Round-Robin algorithm on 1000 different instances. Noted that Algorithm 4 does not have a good performance when the number of jobs is much larger than the number of agents, e.g., the groups with 5 agents (U.1, P.1, N.1) in Figure 1. The reason Algorithm 4 performances unsatisfactorily is that Algorithm 4 stops at the threshold while there are a lot of remaining jobs. To fix this problem, we add the Round-Robin procedure at the end of Algorithm 4, i.e., if there exist some unallocated jobs at the end of Algorithm 4, we run Round-Robin algorithm on the remaining job set. Finally, every agent computes the maximum compatible job set from the union of the job set returned by Algorithm 4 and Round-Robin algorithm. The formal description can be found in Algorithm 10. Let BAG+ be the updated version of Algorithm 4 and BAG be the original one. With the help of the Round-Robin procedure, the performance of Algorithm 4 is better than the Round-Robin algorithm in all groups.

---

**Algorithm 10.** Matching-BagFilling + Round-Robin (BAG+)

---

**Input:** Agent set $A$ and job set $J$.
**Output:** EF1 schedule $\mathbf{X} = (X_1, \cdots, X_m)$
  1: Run Algorithm 4.
  2: Let $\mathbf{X} = (X_1, \cdots, X_m)$ be the schedule returned by Algorithm 4.
  3: Let $X_0 = J \setminus \bigcup_{i \in [m]} X_i$.
  4: **if** $X_0 \neq \emptyset$ **then**
  5:    Run Algorithm 9 with job set $X_0$ and agent set $A$.
  6:    Let $\mathbf{X}' = (X_1', \cdots, X_m')$ be the schedule returned by Algorithm 9.
  7: **end if**
  8: $i = 1$. // The index.
  9: **for all** $X_i$ **do**
 10:    Let $X_i'' \subseteq (X_i \cup X_i')$ be the compatible job set with the maximum weight for agent $a_i$.
 11:    $X_i = X_i''$.
 12: **end for**
 13: $X_0 = J \setminus \bigcup_{i \in [m]} X_i$.

---

According to Figure 1, it is not hard to see that Algorithm 4 is not able to achieve a good performance when the number of jobs is much larger than the number of agents. When the size of the job set is

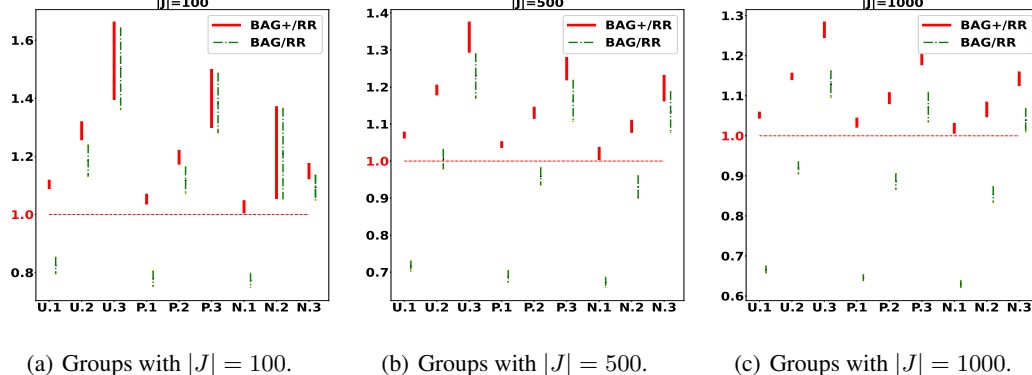

(a) Groups with $|J| = 100$.      (b) Groups with $|J| = 500$.      (c) Groups with $|J| = 1000$.

Figure 1: The results of the evaluation of Algorithm 4, Algorithm 10 and Algorithm 9 on different settings. Every subfigure represents the groups with same job set size. Every notation in x-axis represents a setting. Notations "U.", "P.", "N.", represent the utility gain follows the Uniform, Poisson, Normal Distribution, respectively. Notations "1", "2", "3", represent the number of agents is 5, 10, 15, respectively. The top and bottom point of every solid red interval represent the maximum and minimum value of BAG+/RR among all the agents, where BAG+/RR is the ratio of total gain that the agents receive when we run BAG+ and RR algorithm. The top and bottom point of every dot-dashed green interval represent the maximum and minimum value of BAG/RR among all the agents, where BAG/RR is the ratio of total gain that the agents receive when we run BAG and RR algorithm.

100, Algorithm 4 performs worse than Round-Robin only in the setting where the agent set is 5 (see Figure 1 (a), only U.1, P.1, N.1's green interval is behind 1.0). When we increase the number of jobs to 500, the situation that Algorithm 4 is worse than Round-Robin begins to appear at $|A| = 10$ (see Figure 1 (b), part of green interval of U.2 begins to appear behind 1.0). When we further increase the number of jobs to 1000, Algorithm 4 performs better than Round-Robin only in the setting where there are 15 agents (see Figure 1 (c), only U.3, P.3, N.3's green interval is above 1.0).

The reason is that Algorithm 4 stops at the case where every agent gets the threshold but there are a lot of remaining jobs. We can fix this issue by adding an extra round-robin procedure to allocate the remaining jobs, and thus yield BAG+ algorithm. According to Figure 1, we can find that the performance of BAG+ is better than Round-Robin in all settings as all red intervals are above 1.0. Thus, BAG+ algorithm can achieve a good performance in practices and guarantee the approximation in the worst case.