# OpenReview forum: "Fair Scheduling for Time-dependent Resources"
_NeurIPS.cc/2021/Conference — NeurIPS 2021 Poster_

### Official Review · Reviewer_RGgx · 2021-07-16

**Rating:** 7
**Confidence:** 3

**Summary:**

The paper considers the scheduling problem of assigning jobs, each with a release time a deadline and a duration, to heterogeneous machines with the objective of *fairness*. More specifically, each machine has a utility for each job and its total utility is defined as the sum of utilities of jobs that were assigned to it and can be feasibly scheduled. With respect to fairness two concepts are considered:

- Maximin Share (MMS): the MMS of a machine can be thought of as the value it would obtain in the worst case by partitioning the jobs into bundles and was the last machine to select a bundle. If every machine obtains a utility that is no less than an $\alpha$-fraction of its MMS.

With respect to MMS the paper proves the existence of an $1/3$-approximate MMS schedule and shows that a $(0.24-\epsilon)$ one can be constructed in polynomial time. These improve upon the best known $1/5$-approximate MMS schedule due Ghodsi et al.

-  Envy freeness up to one item (EF1) + Pareto optimal (PO): EF1 means that although envy between different machines may exist, it disappears after removal of one item, and an allocation is PO if there does not exist another allocation that makes nobody worse off but somebody strictly better off.

The authors show that no algorithm can return a schedule satisfying EF1 and PO simultaneously for all input instances. When all durations are unit and the valuations are identical then such a schedule can be computed in polynomial time. It is also shown, that the schedule maximising Nash social welfare is PO and simultaneously 1/4-approximate EF1 (and 1/2-approximate EF1 if all jobs have unit processing times.)

**Ethical Concerns:**

no ethical concerns.

**Limitations And Societal Impact:**

no limitations and no potential negative societal impact that I can think of.

**Main Review:**

Regarding MMS, the algorithm itself is quite involved and consists 1) so-called large jobs are sequentially assigned to the machines so that when a machine receives a job the MMS values of the other machines do not decrease, and 2) the remaining jobs are put into a bundle until some machine chooses to "get" a subset of that bundle. This approach more or less directly gives the existential result but needs to be massaged in order to give the constructive one. I did not check the proofs line by line but they seem to check out.

With respect to EF1 and PO, it is shown that they are not compatible even in special cases. For the special case of unit processing time and identical valuations a polynomial time algorithm is given based on a maximum weighted $m$-matching on a specific graph. Naturally the algorithm crucially builds upon the fact that the jobs have unit sizes. Although the algorithm is only pseudo-polynomial (all possible time-slots are considered individually) it can be transformed into a polynomial one via standard tricks.

In the experimental results the algorithm is compared to -- and outperforms -- Round-Robin.

The paper considers an interesting problem, and at least the result with respect to MMS is interesting since it improves the state of the art, and the algorithm and its analysis are definitely non-trivial (although I cannot say to what extend they build upon previous works and which ideas are new). The results regarding EF1 + PO seem quite specific and not so interesting to me. Overall this is a nice and well written paper and I think it should be accepted, although I am a bit uncertain on whether this is the correct venue for such a result.





**Time Spent Reviewing:**

2.5

---

> ### Author Response · Authors · 2021-08-09
> **Response to Reviewer RGgx**
>
> We sincerely thank the reviewer for the effort in reviewing our submission and are grateful for the positive comments, which are the best encouragement for our work.
>
> We think NeurIPS is the correct venue for our work as it can be seen that NeurIPS is becoming more and more open to Algorithmic Game Theory which is listed in the Call For Papers. Fair division is actually a typical problem in computational social choice, which is one of the four main research areas as shown in wiki (https://en.wikipedia.org/wiki/Algorithmic_game_theory).
>
> Similar works include the following.
>
> (1) Explainable Voting (NeurIPS 2020);
>
> (2) Exploring Algorithmic Fairness in Robust Graph Covering Problems (NeurIPS 2019);
>
> (3) Balancing Efficiency and Fairness in On-Demand Ridesourcing (NeurIPS 2019).

---

### Official Review · Reviewer_kF3C · 2021-07-17

**Rating:** 4
**Confidence:** 4

**Summary:**

This paper presents algorithms for fair division of items to agents. Each item is a job that needs to be processed within a given interval for its utility to be extracted by the agent processing it. This adds a layer of complexity since given any set of items, it is NP-hard to find the subset of feasible items that maximizes utility for the agent (utilities are additive but heterogeneous across agents). Two notions of fairness are considered. The first part of the paper focuses on max-min fairness. The main results are to improve the bounds for (a) the best existential max-min fair allocation, and (b) the best approximation to max-min fairness that can be obtained algorithmically in polynomial time. The difference between the two is primarily because of the NP-hardness stated above of finding the optimal subset of feasible jobs in a given set of jobs. The second paper of the paper focuses on variants of envy freeness. Here, the goal is to obtain envy-free solutions that are also Pareto optimal. But, this is impossible, and so are various extensions such as relaxing to EF1. Nevertheless, some positive results are obtained by relaxing EF1 an approximate notion, or relax Pareto optimality to individual optimality and focus on rigid jobs, etc.

**Ethical Concerns:**

None that I can think of.

**Limitations And Societal Impact:**

Limitations are not discussed explicitly, although the section on future work does point out some ways that the results in this paper can be made more general. I think the paper would benefit from a discussion on the limitations of the fairness measures used here to model fairness as an abstract concept, i.e., what are situations where these are the right measures and when are there other measures that fit fairness goals better?

**Main Review:**

The paper has some interesting and non-trivial results, and as such, and I appreciate the technical work in the paper. But, my main concern is fit. I do not see a clear connection of this paper to ML. It is basically a set of results in designing fair scheduling algorithms. The authors try to explain the relevance by saying that fairness is important in AI and ML. Of course that is true, but this paper does not give any results for fairness in ML. I think the paper is more suitable for a conference like EC, and possibly AAAI, or even some theoretical computer science conferences, but I don't think it is suitable for NeurIPS.

I have read the authors' response. The main concern remains the suitability of the paper for this particular venue, since the connections to learning are still rather vague. On technical merit alone, the score of the paper would be 6 instead of 4, which I have communicated to the AC.

**Time Spent Reviewing:**

1

---

> ### Author Response · Authors · 2021-08-09
> **Response to Reviewer kF3C**
>
> We thank the reviewer for the effort in reviewing our submission and are grateful for the positive words on our results. We will follow your suggestion to discuss the limitations, but we still believe that NeurIPS is a good venue for our work.
>
> Q1:Limitations of the solution concepts
>
> We thank the reviewer for pointing out this point. We will discuss the limitations in the next version. Actually, the solution concepts based on envy-freeness (EF) and those based on maximin share fairness (MMS) have their own merits and limitations.
>
> Regarding EF:
>
> -Merit: It is more stable in the sense that nobody wants to exchange their items with others.
>
> -Limitation: It requires complete information, in the sense that everyone needs to know exactly who gets what, and thus it has privacy issues and is not realistic when the number of items/agents is large (or the agents are distributed in a large social network).
>
> Regarding MMS:
>
> -Merit: Each agent has her own threshold (i.e., MMS_i) which does not depend on any other agent’s allocation or valuation, and thus it does not require complete information.
>
> -Limitation: The allocation may not be as balanced as an EF allocation and it may happen that some agent significantly envies another agent’s bundle.
>
>
> Q2: Whether NeurIPS is the correct venue
>
> We believe NeurIPS is a suitable venue because our work does fall under the umbrella of the track “Theory (e.g., control theory, learning theory, algorithmic game theory)” in Call for Papers. We can see that NeurIPS and ICML are becoming more and more open to Algorithmic Game Theory/Optimization works and other interdisciplinary research, and we believe it is important to know the extent to which fairness can be ensured in any machine-agent system.
>
> 1. Algorithmic game theory and optimization
>
> Fair division is a typical problem in computational social choice, which is one of the four main research areas of algorithmic game theory, as shown in the wiki (https://en.wikipedia.org/wiki/Algorithmic_game_theory). This is because fairness, like Nash equilibrium, is one of the several ways to model the stability of a world. Particularly, the main objective of cooperative game theory is to investigate what kind of fairness can be ensured in a multi-agent system, such as Shapley value and Banzhaf index.
>
> The main focus of our work is to investigate how to (approximately) achieve optimal fairness in a job scheduling setting, and thus also related to the optimization track. Designing (approximation) algorithms to solve optimization problems is a hot topic in NeurIPS/ICML.
>
> These two points can also be verified by the following computational social choice/optimization papers.
>
> (1) Explainable Voting (NeurIPS 2020);
>
> (2) Exploring Algorithmic Fairness in Robust Graph Covering Problems (NeurIPS 2019);
>
> (3) A Graph-Theoretic Additive Approximation of Optimal Transport (NeurIPS 2019).
>
>
>
> 2. Fairness in the (time-dependent) machine-agent system.
>
> How to make fair decisions in machine-agent systems is becoming an important question in NeurIPS/ICML, especially for time-dependent/real-time settings. For example, both of the following works utilize approximation algorithm design approaches to solve the fairness issues in time-dependent settings.
>
> (1) Balancing Efficiency and Fairness in On-Demand Ridesourcing (NeurIPS 2019);
>
> (2) Fairness and Bias in Online Selection (ICML 2021).

---

### Official Review · Reviewer_WBBe · 2021-07-18

**Rating:** 8
**Confidence:** 4

**Summary:**

The paper addresses the following job scheduling problem. There are n jobs and m processors. Job i “arrives” at time r_i, has a processing time p_i, and needs to be executed non-preemptively by time d_i (deadline). It is scheduled on a processor for execution, which can be working on one job at a time and derives utility u_i from execution of a job i. Each processor accumulates the sum of the utilities of the individual jobs it processes.

Two core solution concepts are used for allocating/scheduling the jobs onto processors. 1) maximizing the minimum utility accumulated by any processor, and 2) having the accumulated utility of each processor be no less than that of any other. The latter is matched with a Pareto optimality concept. Variants of those core concepts are investigated.


**Main Review:**

The paper is interesting and provides “relaxation bounds” for construction of schedules in general instances. I can see how the proofs go and actually some of the arguments are reminiscent of those used in classical job-shop scheduling theory. It would be useful to discuss a bit or provide pointers to connections to job-shop scheduling.

Also, it appears from the proofs that if we restrict the classes of “arrival” times r_i, the processing times p_i and deadlines d_i -- so that we reduced the “busrtiness” of the job “flow” – then we can obtain (potentially) much tighter “relaxation bounds” for the performance. That is, it appears that if we control/suppress the “burstiness” of the jobs, we’ll be able to schedule them more efficiently on the same set of processors. I believe the reader would benefit from at least a discussion of that aspect, even if the existing proofs cannot be “massaged” to demonstrate this effect.


**Time Spent Reviewing:**

4.5

---

> ### Author Response · Authors · 2021-08-09
> **Response to Reviewer WBBe**
>
> We thank the reviewer for the positive and constructive comments. We will follow your suggestions and include the corresponding discussions in the next version.
>
> Q1. The connection to Job-shop Scheduling.
>
> First, our Algorithm 3 contains two stages, (i) employing matching (Algorithm 1) to handle the jobs with large value; (ii) and then, using the bag-filling idea to deal with the small jobs. The framework is trying to deal with the hard job first (large jobs for our problem) and then, using another algorithm to deal with the easy job. Treating different types of jobs differently is also used in job shop scheduling. For example, in the paper “Approximation algorithms for flexible job-shop problems”, the authors Klaus Jansen et al. first fix the position of the constant number of long jobs and then employ linear programming to compute the schedule to small jobs. Other than the high-level idea, we do not use similar techniques in each stage due to different settings and objectives.
>
> Second, the drawback of Algorithm 3 is that we need to know the values of every MMS_i, which may need exponential time to compute. To fix this issue, in Algorithm 4, we guess the value of MMS_i from an upper bound, say MMS_i’, and decrease MMS_i’ by a 1/(1+\epsilon) fraction if the resulting allocation cannot satisfy every agent. Guessing (and refining) the optimal value of an NP-hard problem is also used in job shop scheduling, although in a different manner. For example, in the paper “Job Shop Scheduling with Deadlines”, the designed algorithm uses the minimum makespan to guide the scheduling of jobs. To make the algorithm run in polynomial time, they refine the algorithm by guessing an upper bound of the minimum makespan and repeatedly refine the guess via binary search. Note that we cannot use binary search as the correctness of Theorem 2 requires that the guesses are no less than the (MMS_i-\epsilon)’s.
>
> The above are two examples showing the connection between our algorithms and the ones in job shop scheduling. We are not sure whether these are what the reviewer has in mind. We are open to more discussion on this front.
>
> Q2. Tighter bounds on restricted inputs.
>
> We thank the reviewer for pointing out this direction. We agree with the reviewer that when the arrival times, processing times, and deadlines have good features, the approximation ratios of MMS could be (significantly) improved. We will add a discussion about this direction in the next version. For example, if all jobs have the same arrival time and deadline, then the scheduling constraint degenerates to the knapsack constraints (with capacity d_i-s_i), where the processing times can be viewed as jobs’ sizes. The MMS fairness for this setting has not been studied, and our algorithm directly implies 1/3-approximation. But we believe the approximation ratio can be improved. If we further have an infinite deadline, then the scheduling constraint does not have any restrictions anymore. Unfortunately, for this unconstrained setting, the following recent arXiv paper shows that no algorithm can be better than 39/40-approximation.
>
> A tight negative example for MMS fair allocations (arXiv 2021); https://arxiv.org/abs/2104.04977
>
> Thus for any restricted inputs, the approximation ratio is between 39/40 and 1/3.

---

### Official Review · Reviewer_SsyJ · 2021-07-19

**Rating:** 7
**Confidence:** 4

**Summary:**

Problem considered is the following: Given a m intelligent agents or machines and a set of Jobs (with release times and deadlines, processing time requirement and a utility) the goal is to allocate a maximum utility subset of jobs feasibly under some fairness criteria. They consider two fairness criteria namely: Minimax fairness and envy freeness.

The paper obtains novel approximation algorithms for various problems arising in this setting, improving upon the existing best results for some of them.

**Limitations And Societal Impact:**

Quite theoretical - so this question seems N/A

**Main Review:**


I like this paper -- well motivated and well written problem. The algorithm for minimax fairness (1/3-approximation guarantee) which improves upon existing state of the art (1/5-approximation) is in particular quite interesting. The algorithm is surprisingly simple (yet, in my opinion non-trivial to prove) : keep filling a subset ("bag") of tasks until some agent finds it attractive enough. There are other complementary results -- for example making the above algorithm PTIME runnable and those considering the envy free fairness setting.

**Time Spent Reviewing:**

~ 2-3 hours

---

> ### Author Response · Authors · 2021-08-09
> **Response to Reviewer Ssy**
>
> We thank the reviewer for the effort in reviewing our submission and are grateful for the positive comments, which are the best encouragement for our work.

---

### Decision · Program_Chairs · 2021-09-27

**Decision:**

Accept (Poster)

**Comment:**

This paper was well-liked by the reviewers.  The only concern was fit for NeurIPS.  I believe there will be interest in this work and I prefer to error on the side of being inclusive.